# How connectivity structure shapes rich and lazy learning in neural circuits

Yuhan Helena Liu[1,2,3,*], Aristide Baratin[4], Jonathan Cornford[3,5], Stefan Mihalas[1,2], Eric Shea-Brown[1,2], and Guillaume Lajoie[3,6,7,*]

[1]University of Washington, Seattle, WA, USA
[2]Allen Institute for Brain Science, Seattle WA, USA
[3]Mila - Quebec AI Institute, Montreal, QC, Canada
[4]Samsung - SAIT AI Lab, Montreal, QC, Canada
[5]McGill University, Montreal, QC, Canada
[6]Canada CIFAR AI Chair, CIFAR, Toronto, ON, Canada
[7]Université de Montréal, Montreal, QC, Canada
[*]Correspondence: hyliu24@uw.edu, g.lajoie@umontreal.ca

## Abstract

In theoretical neuroscience, recent work leverages deep learning tools to explore how some network attributes critically influence its learning dynamics. Notably, initial weight distributions with small (resp. large) variance may yield a rich (resp. lazy) regime, where significant (resp. minor) changes to network states and representation are observed over the course of learning. However, in biology, neural circuit connectivity could exhibit a low-rank structure and therefore differs markedly from the random initializations generally used for these studies. As such, here we investigate how the structure of the initial weights — in particular their effective rank — influences the network learning regime. Through both empirical and theoretical analyses, we discover that high-rank initializations typically yield smaller network changes indicative of lazier learning, a finding we also confirm with experimentally-driven initial connectivity in recurrent neural networks. Conversely, low-rank initialization biases learning towards richer learning. Importantly, however, as an exception to this rule, we find lazier learning can still occur with a low-rank initialization that aligns with task and data statistics. Our research highlights the pivotal role of initial weight structures in shaping learning regimes, with implications for metabolic costs of plasticity and risks of catastrophic forgetting.

## 1 Introduction

Structural variations can significantly impact learning dynamics in theoretical neuroscience studies of animals. For instance, studies have revealed that specific neural connectivity patterns can facilitate faster learning of certain tasks (Braun et al., 2022; Raman & O'Leary, 2021; Simard et al., 2005; Canatar et al., 2021; Xie et al., 2022; Goudar et al., 2023; Chang et al., 2023). In deep learning, structure, encompassing architecture and initial connectivity, crucially dictates learning speed and effectiveness (Richards et al., 2019; Zador, 2019; Yang & Molano-Mazón, 2021; Braun et al., 2022).

A key structural aspect is the initial connectivity prior to training. Specifically, the initial connection weight magnitude can significantly bias learning dynamics, pushing them towards either rich or lazy regimes (Chizat et al., 2019; Flesch et al., 2021). Lazy learning often induces minor changes in the network during the learning process. Such minimal adjustments are advantageous given that plasticity is metabolically costly (Mery & Kawecki, 2005; Plaçais & Preat, 2013), and significant changes in representations might lead to issues like catastrophic forgetting (McCloskey & Cohen, 1989; Kirkpatrick et al., 2017). On the other hand, the rich learning regime can significantly adapt the network's internal representations to task statistics, which can be advantageous for task feature acquisition and has implications for generalization (Flesch et al., 2021; George et al., 2022). Most research on initial weight magnitude's role in learning dynamics has focused on random Gaussian

or Uniform initializations (Woodworth et al., 2020; Flesch et al., 2021; Braun et al., 2022). These patterns stand in contrast to the connectivity structures observed in biological neural circuits, which could exhibit a more pronounced low-rank eigenstructure (Song et al., 2005). This divergence prompts a pivotal question: how does the initial weight structure, given a fixed initial weight magnitude, bias the learning regime?

This study examines how initial weight structure, particularly the effective rank, modulates the *effective* richness or laziness of task learning within the standard training regime. We note that *rich* and *lazy* learning regimes have well established meanings in deep learning theory. The latter being defined as a situation where the Neural Tangent Kernel (NTK) stays stationary during training, while the former refers to the case where the NTK changes. In this work, we slightly extend these definitions and introduce **effective learning richness/laziness**. Unlike the traditional definition, which is based upon initialization, effective learning richness/laziness is defined in terms of post-training adjustment measurements. From this perspective, a learning process is deemed effectively "lazy" if the measured NTK movement is small. For example, consider a network whose initialization puts it in standard rich regime, but for a given task, its NTK moves very little during training. We define learning for this specific situation as effectively lazy. In other words, while the standard regime definition informs us (prior to training) whether the network can adapt significantly to task training or not, our "effective" definition lies in the post-training effects.

## 1.1 CONTRIBUTIONS

Our main **contributions** and findings can be summarized as follows:

- Through theoretical derivation in two-layer feedforward linear network, we demonstrate that higher-rank initialization results in *effectively* lazier learning **on average** across tasks (Theorem 1). We note that the emphasis of the theorem is on the expectation across tasks.

- We validate our theoretical findings in recurrent neural networks (RNNs) through numerical experiments on well-known neuroscience tasks (Figure 1) and demonstrate the applicability to different initial connectivity structures extracted from neuroscience data (Figure 2).

- We identify scenarios where certain low-rank initial weights still result in *effectively* lazier learning for specific tasks (Proposition 1 and Figure 3). We postulate that such patterns emerge when a neural circuit is predisposed — perhaps due to evolutionary factors or post-development — to certain tasks, ingraining specific inductive biases in neural circuits.

## 1.2 RELATED WORKS

An extended discussion on related works can also be found in Appendix A.

**Theoretical Foundations of Neural Network Regimes and Implications for Neural Circuits:** The deep learning community has made tremendous strides in developing theoretical groundings for artificial neural networks (Advani et al., 2020; Jacot et al., 2018; Alemohammad et al., 2020; Agarwala et al., 2022; Atanasov et al., 2021; Azulay et al., 2021; Emami et al., 2021). A focal point is the 'rich' and 'lazy' learning regimes dichotomy, which have distinct impacts on representation and generalization (Chizat et al., 2019; Flesch et al., 2021; Geiger et al., 2020; George et al., 2022; Ghorbani et al., 2020; Woodworth et al., 2020; Paccolat et al., 2021; Nacson et al., 2022; HaoChen et al., 2021; Flesch et al., 2023). The 'lazy' regime results in minimal weight changes, while the 'rich' regime fosters task-specific adaptations. The transition between these is influenced by various factors, including initial weight scale and network width (Chizat et al., 2019; Geiger et al., 2020).

Deep learning theories increasingly inform studies of biological neural network learning dynamics (Bordelon & Pehlevan, 2022; Liu et al., 2022a; Braun et al., 2022; Ghosh et al., 2023; Saxe et al., 2019; Farrell et al., 2022; Papyan et al., 2020; Tishby & Zaslavsky, 2015). For the rich/lazy regime theory, the existence of diverse learning regimes in neural systems is evident through the resource-intensive plasticity-driven transformations prevalent in developmental phases, followed by more subdued adjustments (Lohmann & Kessels, 2014), and previous investigations characterized neural network behaviors under distinct regimes (Bordelon & Pehlevan, 2022; Schuessler et al., 2023) and discerning which mode yields solutions mimicking neural data (Flesch et al., 2021). Our work extends these studies by examining how initial weight structures affect learning.

**Neural circuit initialization, connectivity patterns and learning:** Extensive research has explored the influence of various random initializations on deep network learning (Saxe et al., 2013; Bahri et al., 2020; Glorot & Bengio, 2010; He et al., 2015; Arora et al., 2019). The literature predominantly focuses on random initialization, but actual neural structures exhibit markedly different connectivity patterns, such as Dale's law and enriched cell-type-specific connectivity motifs (Rajan & Abbott, 2006; Ipsen & Peterson, 2020; Harris et al., 2022; Dahmen et al., 2020; Aljadeff et al., 2015). Motivated by existing evidence of low-rankedness in the brain (Thibeault et al., 2024) and the overrepresentation of local motifs in neural circuits (Song et al., 2005), which could be indicative of low-rank structures due to their influence on the eigenspectrum (Dahmen et al., 2020; Shao & Ostojic, 2023), our study explores the impact of connectivity effective rank on learning regimes. This focus is driven by the plausible presence of such low-rank structures in the brain, potentially revealed through these local motifs. With emerging connectivity data (Campagnola et al., 2022; MICrONS Consortium et al., 2021; Dorkenwald et al., 2022; Winnubst et al., 2019; Scheffer et al., 2020), future work is poised to encompass rich additional features of connectivity.

## 2 SETUP AND THEORETICAL FINDINGS

### 2.1 RNN SETUP

We examine recurrent neural networks (RNNs) because they are commonly adopted for modeling neural circuits (Barak, 2017; Song et al., 2016). We consider a RNN with $N_{in}$ input units, $N$ hidden units and $N_{out}$ readout units (Figure 1A). The update formula for $h_t \in \mathbb{R}^N$ (the hidden state at time $t$) is governed by (Ehrlich et al., 2021; Molano-Mazon et al., 2022):

$$h_{t+1} = \rho h_t + (1 - \rho)(W_h f(h_t) + W_x x_t), \tag{1}$$

where an exponential Euler approximation is made with $\rho = e^{-dt/\tau_m} \in \mathbb{R}$ denoting the leak factor for simulation time step $dt$ and $\tau_m$ denoting the membrane time constant; $f(\cdot) : \mathbb{R}^N \to \mathbb{R}^N$ is the activation function, for which we use $ReLU$; $W_h \in \mathbb{R}^{N \times N}$ (resp. $W_x \in \mathbb{R}^{N \times N_{in}}$) is the recurrent (resp. input) weight matrix and $x_t \in \mathbb{R}^{N_{in}}$ is the input at time step $t$. Readout $\hat{y}_t \in \mathbb{R}^{N_{out}}$, with readout weights $w \in \mathbb{R}^{N_{out} \times N}$, is defined as

$$\hat{y}_t = \langle w, f(h_t) \rangle. \tag{2}$$

The objective is to minimize scalar loss $L \in \mathbb{R}$, for which we use the cross-entropy loss for classification tasks and mean squared error for regression tasks. $L$ is minimized by updating the parameters using variants of gradient descent:

$$\Delta W = -\eta \nabla_W L, \tag{3}$$

for learning rate $\eta \in \mathbb{R}$ and $W = [W_h \quad W_x \quad w^T] \in \mathbb{R}^{N \times (N_{in} + N + N_{out})}$ contains all the trainable parameters. Details of parameter settings can be found in Appendix C.

### 2.2 EFFECTIVE LAZINESS MEASURES

As mentioned above, we introduce *effective* richness and laziness, with effectively lazier (resp. richer) learning corresponding to less (resp. greater) network change over the course of learning. To quantify network change, we adopt the following three measures that have been used previously (George et al., 2022). We note that these measures can be sensitive to other architectural aspects that bias learning regimes, such as network width, so throughout we hold these variables constant when making the comparisons.

**Weight change norm** quantifies the vector norm of change in $W$. Effectively lazier learning should result in a lower weight change norm, and it is quantified as:

$$\|\Delta W\| := \|W^{(f)} - W^{(0)}\|, \tag{4}$$

where $\| \cdot \| = \| \cdot \|_F$; $W^{(0)}$ (resp. $W^{(f)}$) are the weights before (resp. after) training.

**Representation alignment (RA)** quantifies the directional change in a representational similarity matrix (RSM) before and after training. RSM focuses on the similarity between how two pairs

of input are represented by computing the Gram matrix $R$ of last step hidden activity. Greater representation alignment indicates effectively lazier learning in the network, and it is obtained by

$$RA(R^{(f)}, R^{(0)}) := \frac{Tr(R^{(f)}R^{(0)})}{\|R^{(f)}\|\|R^{(0)}\|}, \quad \text{where } R := H^T H, \tag{5}$$

where $H \in \mathbb{R}^{N \times m}$ is the hidden activity at the last time step; $R^{(0)}$ and $R^{(f)} \in \mathbb{R}^{m \times m}$ are the initial and final RSM, respectively; $m$ is the batch size.

**Tangent kernel alignment (KA)** quantifies the directional change in the neural tangent kernel (NTK) before and after training; effectively lazier learning should result in higher tangent kernel alignment. The NTK computes the Gram matrix $K$ of the output gradient. Greater tangent kernel alignment points to effectively lazier learning, and it is obtained by

$$KA(K^{(f)}, K^{(0)}) := \frac{Tr(K^{(f)}K^{(0)})}{\|K^{(f)}\|\|K^{(0)}\|}, \quad \text{where } K := \nabla_W \hat{y}^T \nabla_W \hat{y} \tag{6}$$

where $K^{(0)}$ and $K^{(f)} \in \mathbb{R}^{m \times m}$ (for the $N_{out} = 1$ case) denote the initial and final NTK, respectively.

### 2.3 THEORETICAL FINDINGS

This subsection derives the theoretical impact of initial weight effective rank on tangent kernel alignment. First, Theorem 1 focuses on **task-agnostic** settings, treating task definition as random variables and computing the **expected** tangent kernel alignment across tasks. With some assumptions, tangent kernel alignment is maximized when the initial weight singular values are distributed across all dimensions (i.e. high-rank initialization).

In this section, our theoretical results are framed in a simplified feedforward setting, as we use a two-layer network with linear activations. However, we return to RNNs (Eq. 1) for the rest of the paper, and verify the generality of our theoretical findings with numerical experiments for both feedforward and recurrent architectures. Our choice is motivated by the need for theoretical tractability. While research on RNN learning in the NTK regime exists (Yang, 2020; Alemohammad et al., 2020; Emami et al., 2021), we are not aware of any studies featuring the final converged NTK that could serve as a basis for our comparison of the initial and final kernel. Consequently, we have chosen to focus on RNNs for neural circuit modeling and employ linear feedforward networks for theoretical derivations, a strategy also adopted by Farrell et al. (2022); numerous other studies, including Saxe et al. (2019), (Atanasov et al., 2021), (Arora et al., 2019), and (Braun et al., 2022), have similarly concentrated on extracting theoretical insights from linear feedforward networks.

For a two-layer linear network with input data $X \in \mathbb{R}^{d \times m}$, $W_1 \in \mathbb{R}^{N \times d}$ and $W_2 \in \mathbb{R}^{1 \times N}$ as weights for layers 1 and 2, respectively, the NTK throughout training, $K$, is:

$$K = X^T(W_1^T W_1 + \|W_2\|^2 I)X. \tag{7}$$

Without the loss of generality, suppose the output target $Y \in \mathbb{R}^{1 \times m}$ is generated from a linear teacher network as $Y = \beta^T X$, for some Gaussian vector $\beta \in \mathbb{R}^d$, with $\beta_i \sim \mathcal{N}(0, 1/d)$.

**Theorem 1.** *(Informal) Consider the network above with its corresponding NTK in Eq. 7, trained under MSE loss with small initialization and whitened data. The expected kernel alignment across tasks is maximized with high-rank initialization, i.e. the singular values of $W_1^{(0)}$ are distributed across all dimensions. (Formal statement and proof are in Appendix B)*

The intuition of Theorem 1 result is that, when two random vectors are drawn in high-dimensional spaces, corresponding to the low-rank initial network and the task, the probability of them being nearly orthogonal is very high; this then necessitates greater movement to eventually learn the task direction. We emphasize again that Theorem 1 is **task-agnostic**, i.e. it focuses on the **expected** tangent kernel alignment across input-output definitions. This is in contrast to **task-specific** settings (e.g. Woodworth et al. (2020)) that focus on a given task. In such task-specific settings, certain low-rank initializations can in fact lead to lazier learning. The following proposition predicts that if the task structure is known, low-rank initialization that is already aligned with the task statistics (input/output covariance) can lead to kernel alignment. We revisit this proposition again in Figure 3. We remark that initializing this way can still have high initial error because of randomized $W_2^{(0)}$.

**Proposition 1.** *(Informal) Following the setup and assumptions in Theorem 1, rank-1 initializations of the form $W_1^{(0)} = \sigma[\beta^T/\|\beta\| \quad \vec{0} \quad ... \quad \vec{0}]$ leads to a high tangent kernel alignment. (Formal statement and proof are in Appendix B)*

Above, we state technical results in terms of one metric of the effective laziness of learning — based on the NTK; our proof in Appendix B easily extends also to the representation alignment metric. The impact on weight change is also assessed in Appendix Proposition 2. This is in line with our simulations with RNNs, which will show similar trends for all three of the metrics introduced in Section 2.2).

## 3 SIMULATION RESULTS

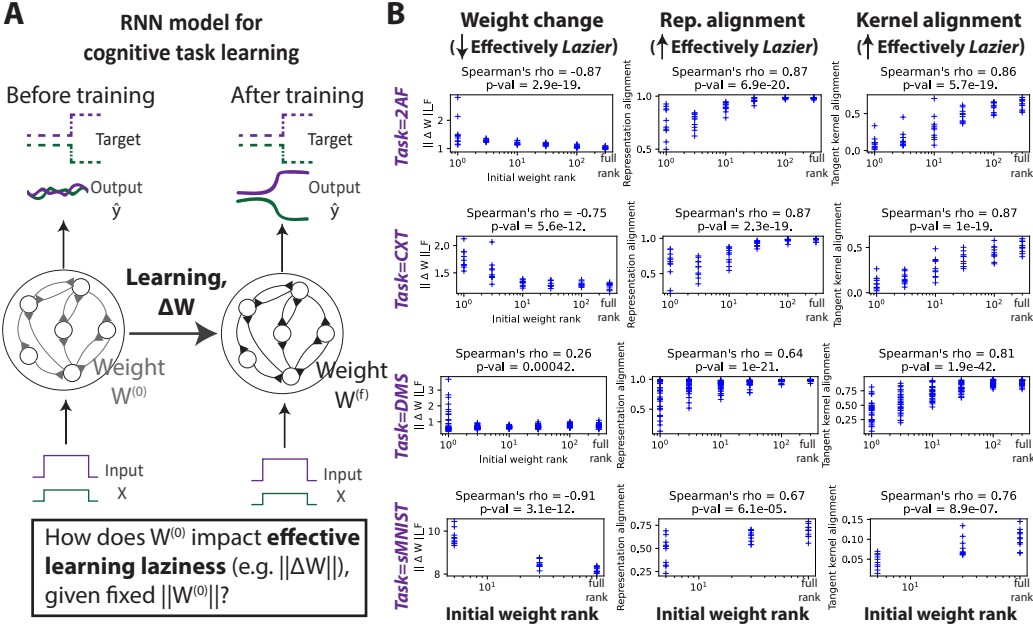

Figure 1: **Low-rank initial recurrent weights, generated using SVD, lead to greater changes (or effectively richer learning) in the recurrent neural network**. A) Schematic of RNN training setup. B) Measurements of effective richness vs laziness of learning (metrics as defined in Section 2.2), for RNN trained on several cognitive tasks in Neurogym (Molano-Mazon et al., 2022) as well as the sequential MNIST task (sMNIST). For details on SVD weight creation, see Appendix C. Fewer rank points were used for sMNIST due to computational time. Each dot represents a single training run, with each run using a different random initialization (10 runs total for each setting).

In this section we empirically illustrate and verify our main theoretical results, which are: **(1)** on average, high-rank initialization leads to effectively lazier learning (Theorem 1); **(2)** it is still possible for certain low-rank initializations that are already aligned to the task statistics to achieve effectively lazier learning (Proposition 1).

**Impact on effective laziness by low-rank initialization via SVD in RNNs:** As a proof-of-concept, we start in Figure 1 with low-rank initialization in RNNs by truncating an initial Gaussian random matrix via Singular Value Decomposition (SVD), which enables us to precisely control the rank, and rescale it to ensure that the comparison is across the same weight magnitude (Schuessler et al., 2020). Additionally, all comparisons were made after training was completed, and all these training sessions achieved comparable losses. For our investigations, we applied this initialization scheme across a variety of cognitive tasks — including two-alternative forced choice (2AF), delayed-match-to-sample (DMS), context-dependent decision-making (CXT) tasks — implemented with Neurogym (Molano-Mazon et al., 2022) and the well-known machine learning benchmark sequential MNIST (sMNIST). Figure 1 indicates that low-rank initial weights result in effectively richer learning and greater network changes.

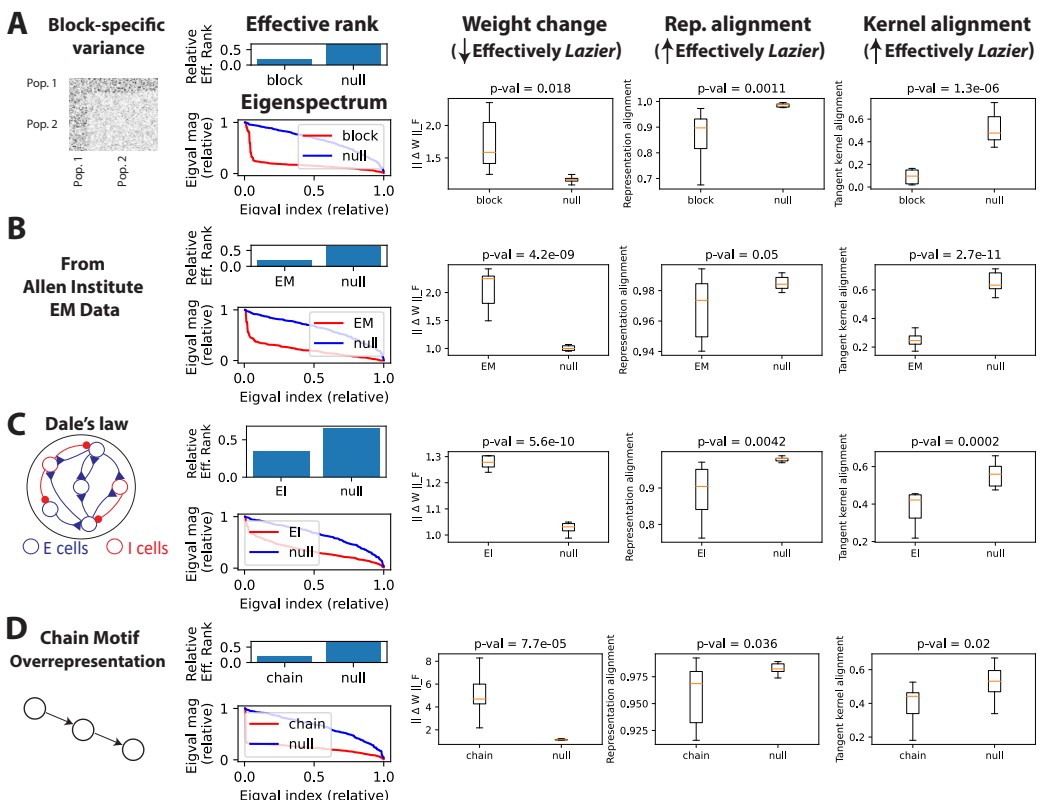

Figure 2: **Low-rank initial weight structures, inspired by biological examples, lead to effectively richer learning**. We present the eigenspectrum and the relative effective rank of connectivity in A) structures with cell-type-specific statistics, B) structures derived from EM data, C) structures obeying Dale's law, and D) structures with an over-representation of chain motifs; we also present the effective learning laziness for networks initialized with these connectivity structures. These structures exhibit a lower effective rank compared to standard random Gaussian initialization (null). We plotted the magnitude of the eigenvalues (Eigval mag) — scaled by the dominant eigenvalue's magnitude — against their indices normalized by the network size $N$ (Eigval index). We apply the effective laziness measures described in Section 2.2 to compare the effective laziness of experimentally-driven initial connectivity versus standard random Gaussian initialization (null). See Appendix C for details on network initialization. The boxplots are generated from 10 independent runs with different initialization seeds. Due to space constraints, we include only the 2AF task here, but Appendix Figures 5 and 6 show that similar trends hold for the DMS and CXT tasks.

These numerical trends are in line with Theorem 1, which focused on an idealized setting of a two-layer linear network with numerical results in Appendix Figure 4A. We also demonstrated this trend for a non-idealized feedforward setting in Appendix Figure 4B, and more explorations in feedforward settings and across a broader range of architecture is left for future exploration due to our focus on RNNs. In the Appendix, we show the main trends observed in Figure 1 also hold for Uniform initialization (Figure 7), soft initial weight rank (Figure 8), various network sizes (Figure 9), learning rates (Figure 10), gains (Figure 11), and finer time step $dt$ (Figure 12). We note that, in addition to fixing the weight magnitude across comparisons, the dynamical regime might also confound learning regimes. A common method for controlling the dynamical regime is through the leading weight eigenvalue, which affects the top Lyapunov exponent. Controlling in this manner led to similar trends (Appendix Figure 13). Investigating the relationship between learning regimes and various concepts of dynamical regimes further is a promising direction for future work. Moreover, since our emphasis is on the effective learning regime, which is based on post-training changes, we concentrated on the laziness measures computed from networks after training, rather than during the learning process. However, we also tracked the alignment with the initial kernel and task kernel alignment during

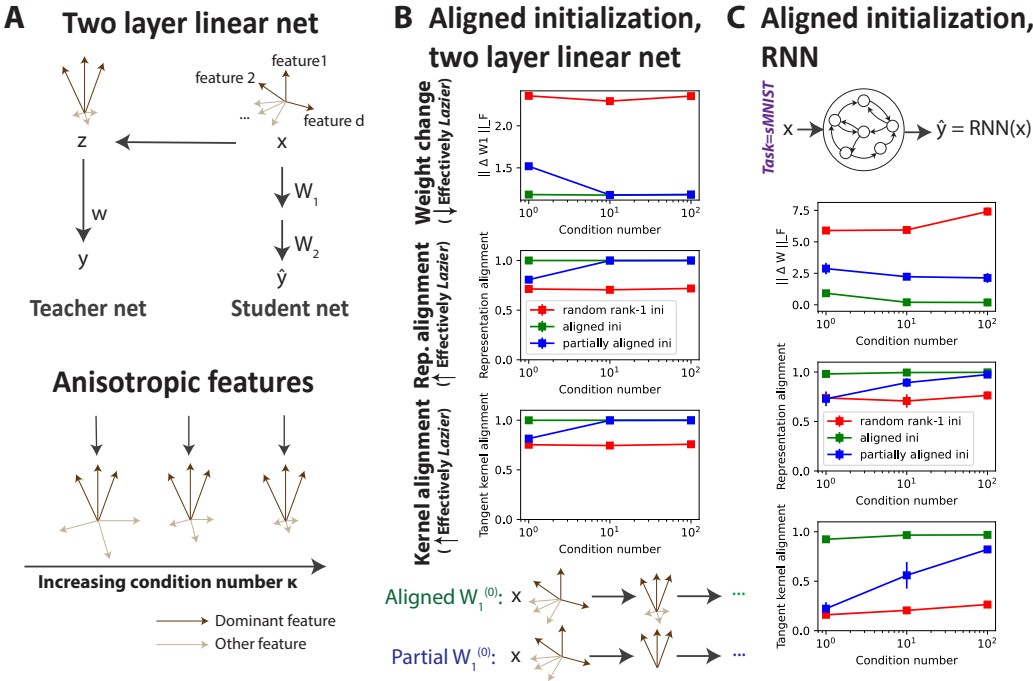

Figure 3: **Low-rank initializations can still achieve high alignment for specific tasks (see Proposition 1).** A) The student-teacher two-layer linear network setup as described in Section 2.3, but with feature anisotropy controlled by a feature modulation matrix $F$, i.e. $z = Fx$. The condition number of $F$ dictates the relative feature strength. We set the top half of the singular values of $F$ are set to $\kappa$, while the bottom half are set to 1, where $\kappa$ represents the condition number of $F$. B) The aligned initialization (green) is achieved by setting $W_1$ as described in Proposition 1 (with $\beta = w^T F$, $w$ is as illustrated), so that the initialization aligns with the task statistics. The partial alignment (blue) mirrors the aligned case, but $F$ is substituted with its rank-$(d/2)$ truncation, causing the network to align only with the dominant features. **We observe that a considerably higher alignment can be achieved when the initialization aligns solely with the dominant features, especially when the relative strength of these dominant features is high.** C) The analysis from B) is replicated for RNNs learning the sMNIST task. As the ground truth network function is elusive, we use a teacher network with pre-trained weights. Once again, we replace $F$ with its rank-$(d/2)$ truncation for partial alignment. Details on the input/output definitions and initializations, as well as other simulation specifics, are available in Appendix C. We note that in all scenarios presented here, the initial errors are high since the readout weights are initialized randomly, rendering it a valid learning problem.

training (Appendix Figure 14). We also examined how the kernel's effective rank evolves throughout the training period (Appendix Figure 15).

**Low-rank initialization via biologically motivated connectivity in RNNs:** To establish a closer connection with biological neural circuits, we have tested our predictions on low-rank initialization using a variety of biologically motivated structures capable of resulting in low-rank connectivity. Here are some of the examples: (A) connectivity with cell-type-specific statistics (Aljadeff et al., 2015), where each block in the weight matrix corresponds to the connections between neurons of two distinct cell types, with the variance of these connections differing from one block to another. In terms of block-specific connectivity statistics, there are infinite possibilities for defining the blocks, each resulting in a unique eigenspectrum. For the example provided here, we adopted the setup from Figure S3 in Aljadeff et al. (2015), with parameters set as $\alpha = 0.02$, $\gamma = 10$, and $1 - \epsilon = 0.8$; these correspond to the fraction of hyperexcitable neurons, gain of hyperexcitable connections and gain of the rest, respectively. We follow this particular setup because it has been demonstrated to create an outlier leading eigenvalue, thereby reducing the effective rank. We also consider (B) connectivity matrix derived from the electron microscopy (EM) data (Allen Institute, 2023), where the synaptic connections between individual neurons are meticulously mapped to create a detailed

and comprehensive representation of neural circuits. Also, we consider (C) connectivity obeying Dale's law, where each neuron is either excitatory or inhibitory, meaning it can only send out one type of signal – either increasing or decreasing the activity of connected neurons – a principle inspired by the way neurons behave in biological systems (Song et al., 2005). Additionally, (D) the over-representation of certain localized connectivity patterns (or network motifs) — such as the chain motif, where two cells are connected via a third intermediary cell — creates outliers in the weight eigenspectrum, subsequently lowering the effective rank (Zhao et al., 2011; Hu et al., 2018; Dahmen et al., 2020). Details of these initial connectivity structures are provided in Appendix C.

As illustrated in Figure 2, these connectivity structures, motivated by known features of biological neural networks, exhibit a lower effective rank compared to standard random Gaussian initialization, thereby serving as natural testbeds for our theoretical predictions. To quantify (relative) effective rank, we used $(\sum_i |\lambda_i|)/(|\lambda_1|N)$, which indicates the fraction of eigenvalues on the order of the dominant one and captures the (scaled) area under the curve of the eigenspectrum plots. We also tried effective rank based on singular values, i.e. $(\sum_i |s_i|)/(|s_1|N)$, in Appendix Figure 16 and observed similar trends. Importantly, Figure 2 show that these different low-rank biologically motivated structures can lead to effectively richer learning compared to the standard random Gaussian initialization. This finding supports our overarching prediction, that lower rank initial weights leads to effectively richer learning. We note that to test our theoretical predictions based on gradient-descent learning without specific constraints on the solutions, the structures are enforced only at initialization and not constrained during training. In Appendix Figure 17, we also constrained Dale's Law throughout training and found similar trends.

**Low-rank initialization aligned with task statistics:** These simulations may be considered to be within our task-agnostic framework. That is, we have chosen a "random" battery of tasks that is not directly matched to the initial network connectivity structures. Thus, our findings that lower rank initializations lead to richer learning are expected from our theoretical prediction on the task-averaged alignment (Theorem 1), rather than something task-specific. However, Proposition 1 also predicts that low-rank initialization can lead to lazy learning if the initialization is already aligned to the task structure. To test this, we observe in Figure 3 that a considerably higher alignment can be achieved when the initialization aligns solely with the dominant task features, especially when the relative strength of these dominant features is high. We postulate that such alignment may occur in biological settings if the circuit has evolved to preferentially learn specific tasks.

## 4 DISCUSSION

Our investigation casts light on the nuanced influence of initial weight effective rank on learning dynamics. Anchored by Theorem 1, our theoretical findings underscore that high-rank random initialization generally facilitates *effectively* lazier learning on average across tasks. This focus on the expectation across tasks can provide insights into the circuit's flexibility in learning across a broad range of tasks as well as predict the effective learning regime when the task structure is uncertain. However, certain low-rank initial weights, when naturally predisposed to specific tasks, may lead to effectively lazier learning, suggesting an interesting interplay between evolutionary or developmental biases and learning dynamics (Proposition 1). Our numerical experiments on RNNs further validate these theoretical findings illustrating the impact of initial rank in diverse settings.

**Potential implications to neuroscience:** We investigate the impact of effective weight rank on learning regimes due to its relevance in neuroscience. Learning regimes reflect the extent of change through learning, implicating metabolic costs and catastrophic forgetting (McCloskey & Cohen, 1989; Plaçais & Preat, 2013; Mery & Kawecki, 2005). The presence of different learning regimes is demonstrated in neural systems, since during developmental phases where neural circuits undergo extensive, plasticity-driven transformations. In contrast, mature neural circuits exhibit more subtle synaptic adjustments (Lohmann & Kessels, 2014). We hypothesize that a circuit's task-specific alignment might be established either evolutionarily or during early development. The specialization of neural circuits, such as ventral versus dorsal (Bakhtiari et al., 2021), may arise from engaging in tasks with similar computational demands. Conversely, circuits with high-rank structures may be less specialized, handling a wider array of tasks. Our framework could be used to compare connectivities across brain regions and species in order to predict their function and flexibility, assessing their functional specialization based on effective connectivity rank. Additionally, our framework predicts

that connectivity rank will affect the degree of change in neural activity during the learning of new tasks. This hypothesis could be tested through BCI experiments, as shown in Sadtler et al. (2014) and Golub et al. (2018), to explore how learning dynamics vary with connectivity rank.

Regarding deep learning, low-rank initialization is not a common practice, yet adaptations like low-rank updates have gained popularity in training large models (Hu et al., 2021). LoRA, the study cited, concentrates on parameter updates rather than initializations, but understanding how update rank affects learning regimes is crucial. Our results offer a starting point for further exploration in this area. Although different rank initializations are less explored, with some exceptions like Vodrahalli et al. (2022), our findings suggest that this area should receive more attention due to its potential effects on learning regimes and, consequently, on generalization (George et al., 2022).

**Limitations and future directions:** Our study predominantly focused on the weight (effective) rank, leaving the exploration of other facets of weight on the effective learning regime as an open avenue. Also, the ramifications of effective learning regimes on learning speed — given the known results on kernel alignment and ease of learning (Bartlett et al., 2021) and present mixed findings in the existing literature (Flesch et al., 2021; George et al., 2022) — warrant further exploration.

Expanding the scope of our study calls for examining a wider variety of tasks, neural network architectures, and learning rules. Although our work is based on the backpropagation learning rule, its implications for biologically plausible learning rules remain unexplored. Our primary criterion for selecting tasks was their relevance to neuroscience, aligning with our main objectives. However, given the diverse range of tasks performed by various species, future research could benefit from exploring a more extensive array of tasks. Exploring more complex neuroscience tasks, such as those in Mod-Cog (Khona et al., 2023), could provide valuable insights. On that note, we tested the pattern generation task from Bellec et al. (2020), a neuroscience task differing in structure from the Neurogym tasks, and observed similar trends (refer to Appendix Figure 18).

Additionally, we ensured the consistency of outcomes against factors like width, learning rate, and initial gain (see Appendix D), but other factors such as dynamical regime and noise (HaoChen et al., 2021) remain underexamined. On that note, the study's focus on RNNs with finite task duration prompts further investigation into the implications for tasks with extended time steps and how conclusions for feedforward network depth (Xiao et al., 2020; Seleznova & Kutyniok, 2022) translate to RNN sequence length. Examining several mechanisms at once is beyond the scope of one paper, but our theoretical work constitutes the foundation for future investigations.

Moreover, it is crucial to further explore the neuroscientific implications of effective learning regimes, as well as their diverse impacts on aspects such as representation, including kernel-task alignment (see Appendix Figure 14), and generalization capabilities (Flesch et al., 2021; George et al., 2022; Schuessler et al., 2023). Our current study did not delve into how initial weight rank affects these facets of learning, representing an essential future direction in connecting weight rank to these theoretical implications in both biological and artificial neural networks.

Furthermore, while there exists evidence for low-rankedness in the brain (Thibeault et al., 2024), the extent to which the brain uses low-rank structures remains an open question, especially as neural circuit structures can vary across regions and species. While local connectivity statistics (Song et al., 2005) can offer some predictive insight into the global low-rank structure, this relationship is not always immediately apparent (Shao & Ostojic, 2023). Our theoretical results contribute to understanding the role of connectivity rank in the brain by linking effective connectivity rank with learning dynamics.

Lastly, we have primarily examined low-rank tasks and there remains unexplored terrain regarding the interplay between the number of task classes and weight rank, which is pivotal to uncovering a more precise relationship between the effective learning regime and the initial weight rank (Dubreuil et al., 2022; Gao et al., 2017). Overall, this dynamic area of learning regimes is ripe for many explorations, integrating numerous factors; our work contributes to this exciting area with new tools.

## 5 ACKNOWLEDGEMENT

We thank Andrew Saxe, Stefano Recanatesi, Kyle Aitken and Dana Mastrovito for insightful discussions and helpful feedback. This research was supported by NSERC PGS-D (Y.H.L.); FRQNT B2X

(Y.H.L.); Pearson Fellowship (Y.H.L.); NSF AccelNet IN-BIC program, Grant No. OISE-2019976 AM02 (Y.H.L.); NIH BRAIN, Grant No. R01 1RF1DA055669 (Y.H.L., E.S.B., S.M.); Mitacs Globalink Research Award (Y.H.L.); IVADO Postdoctoral Fellowship (J.C); the Canada First Research Excellence Fund (J.C.); NSERC Discovery Grant RGPIN-2018-04821 (G.L); Canada Research Chair in Neural Computations and Interfacing (G.L.); Canada CIFAR AI Chair program (G.L.). We also thank the Allen Institute founder, Paul G. Allen, for his vision, encouragement, and support.

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
