## A EXTENDED DISCUSSIONS ON RELATED WORKS

**Theoretical Foundations of Neural Network Regimes and Implications for Neural Circuits:**
The journey of understanding deep learning systems has borne witness to unprecedented progress
in the mathematical dissection of neural network functionalities (Advani et al., 2020; Jacot et al.,
2018; Pezeshki et al., 2021; Baratin et al., 2021; Alemohammad et al., 2020; Yang, 2020; Agarwala
et al., 2022; Atanasov et al., 2021; Azulay et al., 2021; Emami et al., 2021). These theoretical
findings, until recently confined predominantly to artificial domains, have embarked upon explorations
into biological neural networks, elucidating the intricate dynamics of learning and computational
properties (Bordelon & Pehlevan, 2022; Liu et al., 2022a; Braun et al., 2022; Ghosh et al., 2023).
Among the vanguard of these theoretical endeavors stands the dichotomy of 'rich' and 'lazy' learning
regimes. Both lead to task learning, yet they carry distinct ramifications for representation and
generalization (Chizat et al., 2019; Flesch et al., 2021; Geiger et al., 2020; George et al., 2022;
Ghorbani et al., 2020; Woodworth et al., 2020; Paccolat et al., 2021; Nacson et al., 2022; HaoChen
et al., 2021; Flesch et al., 2023). In the 'lazy' regime, which is typically associated with large
initial weights, learning predominantly centers on adjusting the readout weights. This leads to
minimal alterations in the network weights and representation, while capitalizing on the expansive
dimensionality provided by the hidden layer's random projections (Flesch et al., 2021). In contrast,
the 'rich' regime, defined by smaller initial weights, fosters the development of highly tailored hidden
unit representations specifically aligned with task demands, resulting in considerable adaptations in
weights and representation. It's essential to highlight that the transition and dominance between these
regimes are influenced by more than just the initial weight scale. Other factors, ranging from network
width to the output gain (often referred to as the $\alpha$ parameter), play a pivotal role (Chizat et al., 2019;
Geiger et al., 2020).

A nexus between deep learning theoretical frameworks and neuroscience has unveiled applications of
the rich/lazy regimes. Previous investigations characterized neural network behaviors under distinct
regimes (Bordelon & Pehlevan, 2022; Schuessler et al., 2023) and discerning which mode yields
solutions mimicking empirical data (Flesch et al., 2021). It is compelling to observe that the existence
of multiple learning regimes isn't an isolated phenomenon in artificial systems; analogous learning
patterns echo in neural circuits as well. For instance, while plasticity-driven transformations might
be resource-intensive, they manifest robustly during such developmental phases, followed by minor
changes afterwards (Lohmann & Kessels, 2014). Building upon these findings, our research delves
deeper into the precursors of these regimes. We examine how inherent factors in the brain, especially
initial weight configurations, influence the inclination towards either rich or lazy learning. This
understanding is crucial for assessing the applicability of regime-specific tools in neural contexts and
for shedding light on the potential benefits of having both learning regimes coexist in the brain.

**Interplay of Neural Learning and structure:** Understanding how the brain learns using its myriad
elements is a perennial quest in neuroscience. Addressing this, certain studies have unveiled biologi-
cally plausible learning rules (Lillicrap et al., 2020; Scellier & Bengio, 2017; Diederich & Opper,
1987; Hinton, 2022; Laborieux & Zenke, 2022; Greedy et al., 2022; Sacramento et al., 2018; Payeur
et al., 2021; Roelfsema & Holtmaat, 2018; Meulemans et al., 2022; Murray, 2019; Bellec et al., 2020;
Liu et al., 2021; 2022b; Marschall et al., 2020), suggesting potential neural algorithms involving
known neural ingredients. Concurrently, given the three primary components of a neural network's
design — task, learning rule, and architecture — another avenue of research delves deep into the
architectural facet, specifically focusing on how it interacts with the learning rule to enhance learning
(Richards et al., 2019; Zador, 2019; Yang & Molano-Mazón, 2021). Under the structural umbrella,
the neural unit's complexity and initial connectivity patterns are two crucial aspects. Complex neuron
models, for instance, have shown the potential in boosting learning performance by allowing implicit
forms of memory and computations at the single neuron level (Salaj et al., 2021; Winston et al.,
2023). Moreover, A large body of work has investigated the effect of different random initializations
on learning in deep networks (Saxe et al., 2013; Bahri et al., 2020; Glorot & Bengio, 2010; He
et al., 2015; Arora et al., 2019). For instance, the variance in random initial weights can induce
pronounced shifts in network behavior, ranging from the "lazy" to the "rich" regimes (Chizat et al.,
2019; Flesch et al., 2021). This introduces unique inductive biases during the learning process, with
distinct preferences for learning certain features (George et al., 2022). Our discourse primarily orbits
around connectivity and its implications on learning dynamics in networks with simple rectified units.
Our results sit within the purview of these regimes, with a widely adopted assumption of gradient

descent via backpropagation as the learning rule, while remaining open to encompassing a wider spectrum of rules in future explorations.

**Neural circuit connectivity pattern and eigenspectrum:** While the importance of initial weights on function and learning is clear, the impact of specific weight shapes, apart from weight scale, on rich or lazy dynamics remains less explored. The predominant focus in the literature has been on random initialization. Yet, neural circuit structures significantly diverge from this paradigm. Illustratively, one finds connectivity principles or patterns markedly different from what one observes with a mere random initialization (Pogodin et al., 2023), resulting in distinct neural dynamics; these connectivity principles or patterns include Dale's law (Rajan & Abbott, 2006; Ipsen & Peterson, 2020; Harris et al., 2022), an over-representation of higher-order motifs (Dahmen et al., 2020) and cell-type-specific connectivity statistics (Aljadeff et al., 2015), to name a few. Given the prominence of low-rankedness observed in neural circuits (Song et al., 2005), our study centers on the influence of effective rank on the effective learning regime. As the next generation of connectivity data becomes available (Campagnola et al., 2022; MICrONS Consortium et al., 2021; Dorkenwald et al., 2022; Winnubst et al., 2019; Scheffer et al., 2020), future explorations will broaden the scope to other weight characteristics.

## B PROOFS

### B.1 PROOFS FOR MAIN TEXT THEOREM AND PROPOSITION

**Notation** Let $f(x) = W_2 W_1 x$ denote a two-layer linear network with $N$ hidden units on $d$-dimensional inputs $x \in \mathbb{R}^d$, with weight matrices $W_1 \in \mathbb{R}^{N \times d}$ and $W_2 \in \mathbb{R}^{1 \times N}$. We consider $m$ training inputs $x_1, \cdots x_m$ and the corresponding data matrix $X = [x_1^T \cdots x_m^T] \in \mathbb{R}^{d \times m}$; the output target is generated from a linear teacher network as $Y = \beta^T X$, where $\beta_i \sim \mathcal{N}(0, 1/d)$.

Since our goal is to investigate how the *shape* of the initial weights impacts network change, we will consider a fixed small (Froebenius) norm for these; i.e.,

$$\|W_1^{(0)}\|_F = \|W_2^{(0)}\|_F := \sigma \ll 1$$

We denote by $s_1, \cdots, s_d$ denote the singular values of $W_1^{(0)}$; they satisfy $\sum_{j=1}^d s_j^2 = \sigma^2$.

In what follows we focus on the whitened setting, where $X$ has all its non zero singular values equal to 1. We also assume $m \geq d$ for simplicity (this assumption can easily be relaxed in our analysis), so that the whitened data assumption translates as $XX^T = I_d$.

**Prior results** Our analysis builds on prior results Atanasov et al. (2021) on the evolution of the NTK for two-layer linear networks trained by gradient flow of the mean square error. In the above setting, Atanasov et al. (2021) show that the final NTK $K^{(f)}$ (i.e. the asymptotic NTK as the number of iterations goes to infinity) is given by

$$K^{(f)} = \|\beta\| X^T (\hat{\beta}\hat{\beta}^T + I_d) X + O(\sigma^2). \tag{8}$$

where $\hat{\beta} := \beta/\|\beta\|$. We are interested in the expected kernel alignment over the tasks, in the small initialization regime:

$$\mathbb{E}_\beta[KA(K^{(f)}, K^{(0)})] := \mathbb{E}_\beta \left[ \frac{\text{Tr}(K^{(f)} K^{(0)})}{\|K^{(f)}\|_F \|K^{(0)}\|_F} \right]. \tag{9}$$

**Theorem 1.** *In the above setting, when considering all possible initializations $W_1^{(0)}$ with small fixed norm $\sigma$, the expected kernel alignment $\mathbb{E}_\beta[KA]$ (defined in Eq. 9) is maximized with high-rank isotropic initialization, i.e with $W_1^{(0)}$ that has all its non-zero singular values equal in absolute value.*

*Proof.* Let us write $K^{(0)} = X^T M_0 X$ with $M_0 := W_1^{(0)T} W_1^{(0)} + \sigma^2 I_d$. Up to $O(\sigma^4)$ terms, the numerator in Eq. 9 takes the form

$$
\begin{aligned}
\text{Tr}(K^{(f)} K^{(0)}) &= \|\beta\| \text{Tr}(X^T (\hat{\beta}\hat{\beta}^T + I_d) X X^T M_0 X) \\
&\overset{(a)}{=} \|\beta\| \text{Tr}(X^T (\hat{\beta}\hat{\beta}^T + I_d) M_0 X) \\
&\overset{(b)}{=} \|\beta\| \text{Tr}((\hat{\beta}\hat{\beta}^T + I_d) M_0 X X^T) \\
&\overset{(a)}{=} \|\beta\| \text{Tr}((\hat{\beta}\hat{\beta}^T + I_d) M_0) \\
&\overset{(c)}{=} \|\beta\|(\hat{\beta}^T M_0 \hat{\beta} + \text{Tr } M_0)
\end{aligned}
\tag{10}
$$

where $(a)$ uses $XX^T = I_d$, $(b)$ the cyclicity of the trace, and $(c)$ the fact that $\hat{\beta}^T M_0 \hat{\beta}$ is a scalar.

As for the denominator in Eq. 9), we have,

$$
\begin{aligned}
\|K^{(0)}\|_F^2 &= \text{Tr}(K^{(0)} K^{(0)}) \\
&= \text{Tr}(X^T M_0 X X^T M_0 X) \\
&\overset{(a)}{=} \text{Tr}(M_0^2)
\end{aligned}
\tag{11}
$$

and, up to $O(\sigma^4)$ terms,

$$
\begin{aligned}
\|K^{(f)}\|_F^2 &= \mathrm{Tr}(K^{(f)}K^{(f)}) \\
&= \|\beta\|^2 \, \mathrm{Tr}(X^T(\hat{\beta}\hat{\beta}^T + I_d)XX^T(\hat{\beta}\hat{\beta}^T + I_d)) \\
&= \|\beta\|^2 \, \mathrm{Tr}(X^T(\hat{\beta}\hat{\beta}^T + I_d)X) \\
&\overset{(a)}{=} \|\beta\|^2 \, \mathrm{Tr}(\hat{\beta}\hat{\beta}^T + I_d)^2 \\
&\overset{(b)}{=} \|\beta\|^2(d+3)
\end{aligned}
\tag{12}
$$

where $(a)$ in these two calculations uses $XX^T = I_d$ and the cyclicity of the trace; and $(b)$ notes that the $d \times d$ matrix $\hat{\beta}\hat{\beta}^T + I_d$ has $d-1$ eigenvalues equal to 1 and one equal to 2. Eq. 11 and 12 yield

$$
\|K^{(f)}\|_F \|K^{(0)}\|_F = \|\beta\|\sqrt{(d+3)\,\mathrm{Tr}\,M_0^2}
\tag{13}
$$

Putting together Eq. 10, 13 , we obtain, up to additive $O(\sigma^2)$ terms,

$$
\mathrm{KA}(K^{(f)}, K^{(0)}) = \frac{\hat{\beta}^T M_0 \hat{\beta} + \mathrm{Tr}\,M_0}{\sqrt{(d+3)\,\mathrm{Tr}\,M_0^2}}
\tag{14}
$$

Next, averaging over the tasks requires computing the Gaussian average

$$
A[M_0] := \mathbb{E}_\beta\left[\hat{\beta}^T M_0 \hat{\beta}\right] = \mathbb{E}_\beta\left[\frac{\beta^T M_0 \beta}{\|\beta\|^2}\right].
$$

**Lemma 1.** *The map $A$ is invariant under the action of the orthogonal group, i.e $A[UMU^T] = A[M]$ for all $M \in \mathbb{R}^{d \times d}$ and all orthogonal matrices $U \in \mathbb{R}^{d \times d}$.*

*Proof.* This is a consequence of the invariance of the Gaussian measure under the action of the orthogonal group. Explicitly, given an orthogonal matrix $U$,

$$
\begin{aligned}
A[UMU^T] &= \frac{1}{(2\pi d)^{d/2}}\int \mathrm{d}^d\beta\, e^{-\|\beta\|^2/d}\left[\frac{\beta^T UMU^\top \beta}{\|\beta\|^2}\right] \\
&\overset{\beta':=U^T\beta}{=} \frac{1}{(2\pi d)^{d/2}}\int \mathrm{d}^d\beta'\,|\det U|e^{-\|U\beta'\|^2/d}\left[\frac{\beta'^T M\beta'}{\|U\beta'\|^2}\right] \\
&= \frac{1}{(2\pi d)^{d/2}}\int \mathrm{d}^d\beta'\, e^{-\|\beta'\|^2/d}\left[\frac{\beta'^T M\beta'}{\|\beta'\|^2}\right] \\
&= A[M]
\end{aligned}
\tag{15}
$$

where the third equality follows from $|\det U| = 1$ and $\|U\beta\| = \|\beta\|$. □

**Lemma 2.** *There is a constant $c$ such that $A[M] = c\,\mathrm{Tr}(M)$ for any symmetric matrix $M$.*

*Proof.* Given a symmetric matrix $M$, it can be diagonalized as $M = UDU^T$ where $D = \mathrm{Diag}(\mu_1, \cdots \mu_d)$ is diagonal and $U$ is orthogonal. By rotation invariance from Lemma 1, we have $A[M] = A[D]$, and

$$
A[D] = \mathbb{E}_\beta\left[\hat{\beta}^T D\hat{\beta}\right] = \mathbb{E}_\beta\left[\sum_{j=1}^d \hat{\beta}_j^2 \mu_j\right] = \sum_{j=1}^d \mathbb{E}_\beta\left[\frac{\beta_j^2}{\|\beta\|^2}\right]\mu_j := \sum_{j=1}^d c_j\mu_j
\tag{16}
$$

We conclude by noting that, by invariance of the (isotropic) Gaussian measure under permutation of the vector components, the coefficients $c_j$ are independent of $j$, i.e $c_j \equiv c$ for all $j$. In sum,

$$
A[M] = A[D] = c\,\mathrm{Tr}\,D = c\,\mathrm{Tr}\,M.
\tag{17}
$$

□

The expected kernel alignment thus takes the form,

$$\mathbb{E}_\beta[\text{KA}(K^{(f)}, K^{(0)})] = \frac{(1+c)\,\text{Tr}\,M_0}{\sqrt{(d+3)\,\text{Tr}\,M_0^2}} \tag{18}$$

up to additive $O(\sigma^2)$ terms. Finally, we note that

$$\begin{aligned}
\text{Tr}\,M_0 &= \text{Tr}(W_1^{(0)T}W_1^{(0)} + \sigma^2 I_d) \\
&= \|W_1^{(0)}\|_F^2 + d\sigma^2 \\
&= (d+1)\sigma^2
\end{aligned} \tag{19}$$

and

$$\begin{aligned}
\text{Tr}\,M_0^2 &= \text{Tr}(W_1^{(0)T}W_1^{(0)} + \sigma I_d)^2 \\
&= \sum_{j=1}^d (s_j^2 + \sigma^2)^2 \\
&= \sum_{j=1}^d s_j^4 + 2\sigma^2 \sum_{j=1}^d s_j^2 + d\sigma^4 \\
&= \sum_{j=1}^d s_j^4 + (d+2)\sigma^4
\end{aligned} \tag{20}$$

Substituting into Eq. 18, we have, up to additive $O(\sigma^2)$ terms,

$$\mathbb{E}_\beta[\text{KA}(K^{(f)}, K^{(0)})] = \frac{(1+c)(d+1)}{\sqrt{(d+3)(d+2+\sum_{j=1}^d (s_j/\sigma)^4)}} \tag{21}$$

Finally, we see in Eq 21 that the maximization of $\mathbb{E}_\beta[KA]$ reduces to the following convex constrained optimization problem:

$$\min_s \sum_j s_j^4, \quad \text{subject to} \sum_j s_j^2 = \sigma^2. \tag{22}$$

The KKT solutions satisfy $s_i^2 = \sigma^2/d$ for all $j = 1 \cdots d$. This implies that the expected tangent kernel alignment is maximized when the initial weight singular values $|s_i|$ are distributed evenly across dimensions, which corresponds to a high-rank initialization. □

**Proposition 1.** *Following the setup and assumptions in Theorem 1, rank-1 initialization with $W_1^{(0)} = \sigma[\hat{\beta}^T \quad \vec{0} \quad \dots \quad \vec{0}]$ leads to maximal aligment, i.e, $\text{KA}(K^{(f)}, K^{(0)}) = 1$ up to additive $O(\sigma^2)$ terms.*

*Proof.* We indeed have,

$$\begin{aligned}
K^{(0)} &= X^T(W_1^{(0)T}W_1^{(0)} + \|W_2^{(0)}\|^2 I)X \\
&= \sigma^2 X^T(\hat{\beta}\hat{\beta}^T + I)X
\end{aligned} \tag{23}$$

Thus, writing $K := X^T(\hat{\beta}\hat{\beta}^T + I)X$ and using Eq. 8, the alignment takes the form

$$\begin{aligned}
\text{KA}(K^{(f)}, K^{(0)}) &:= \frac{\text{Tr}(K^{(f)}K^{(0)})}{\|K^{(f)}\|_F \|K^{(0)}\|_F} \\
&= \frac{\text{Tr}(K(K + O(\sigma^2)))}{\|K\|_F \|K + O(\sigma^2)\|_F} \\
&= \frac{\text{Tr}(K^2)}{\|K\|_F^2} + O(\sigma^2) \\
&= 1 + O(\sigma^2)
\end{aligned} \tag{24}$$

□

## B.2 LEARNING REQUIREMENT BASED ON $W_h^{(0)}$ RANK

The focus of this idea is to show that no changes to hidden weights $W_h$ is not possible (e.g. reservoir settings) for zero-error when the initial weight rank falls below a certain threshold. Freezing the hidden weights $W_h$ would be a special case of lazy learning.

**Proposition 2.** *Consider a linear RNN with input at time $t$ as $X_t \in \mathbb{R}^{N \times d}$ (for $t = 1, ..., T - 1$), target output $Y \in \mathbb{R}^{N_{out} \times d}$ only at the last step, recurrent weight matrix $W_h \in \mathbb{R}^{N \times N}$ and readout weight matrix $w \in \mathbb{R}^{N_{out} \times N}$. Here, $N, N_{out}, d$ and $T$ are the number of hidden units, number of classes, number of data points and number of time steps, respectively, and we assume $N, d > N_{out}$. Define initial recurrent weight $W_h^{(0)}$ and final recurrent weight $W_h^{(f)}$ that achieves zero error. Then, for arbitrary input $X$ and target output $Y$, $W_h^{(f)} = W_h^{(0)}$ is not possible when $rank(W_h^{(0)}) < N_{out}$.*

*Proof.* We have the following based on the assumption of the RNN structure, if zero-error learning is achieved:

$$Y = w^{(f)} W_h^{(f)} \left( \sum_{t=1}^{T-1} W_h^{(f)^{T-t-1}} X_t \right). \tag{25}$$

We can prove by contradiction. Suppose $W_h^{(f)} = W_h^{(0)}$, then

$$Y = w^{(f)} W_h^{(0)} \left( \sum_{t=1}^{T-1} W_h^{(0)^{T-t-1}} X_t \right). \tag{26}$$

Since $Y$ is arbitrary, we can have $rank(Y) = N_{out}$ (by the assumption of $N, d > N_{out}$). Applying $rank(W_h^{(0)}) < N_{out}$ we have

$$rank(Y) = rank(w^{(f)} W_h^{(0)} \left( \sum_{t=1}^{T-1} W_h^{(0)^{T-t-1}} X_t \right))$$
$$\overset{(a)}{\leq} min(rank(w^{(f)}), rank(W_h^{(0)}), \left( \sum_{t=1}^{T-1} W_h^{(0)^{T-t-1}} X_t \right))$$
$$< N_{out}, \tag{27}$$

where $(a)$ is because $rank(W_h^{(0)}) < N_{out}$ so the minimum has to be less than $N_{out}$. This would contradict an arbitrary $Y$ with $rank(Y) = N_{out}$. Thus, $W_h^{(f)} = W_h^{(0)}$ cannot happen and recurrent weights have to be adjusted. □

## C  SETUP AND SIMULATION DETAILS

### C.1  INITIAL LOW-RANK WEIGHTS CREATION

For the null case, we initialized with random Gaussian distributions where each weight element $W_{ij} \sim \mathcal{N}(0, g^2/N)$, with an initial weight variance of $g$. Unless otherwise mentioned, we set $g = 1.5$ and network size $N = 300$, though we also validated across other parameter choices (see Appendix D). Input and readout weights were initialized similarly as in Yang & Wang (2020) (see their $EIRNN.ipynb$ notebook).

To create low-rank weights using SVD, we generated temporary weights $\hat{W}_{ij} \sim \mathcal{N}(0, g^2/N)$. Subsequently, we applied SVD to $\hat{W}$ and retained the top components based on the desired rank. To ensure comparisons are made across constant initial weight magnitudes, the resultant weight matrix was rescaled to match the Frobenius norm of $\hat{W}$.

Furthermore, we present details for experimentally-driven low-rank weights. For block-specific statistics, we followed the setup in Figure S3 of Aljadeff et al. (2015), setting parameters as $\alpha = 0.02$, $\gamma = 10$, and $1 - \epsilon = 0.8$. These parameters substantially influence the weight eigenspectrum, as depicted in Figure S3 of Aljadeff et al. (2015); we selected these values specifically to emphasize the outliers and achieve a lower effective rank. These parameters represent the fraction of hyperexcitable neurons (population 1), gain of hyperexcitable connections, and the gain of remaining connections, respectively. For the creation of a chain motif, we employed the procedure described in Section S3.10 of Dahmen et al. (2020), setting $\tau_{chn} = 0.03$ (and $\tau_{chn} = -0.1$ for over-representation or under-representation of the chain motif, respectively). Here, we set $N = 100$. These parameters were chosen to provide enough distinctions from the null case, while still ensuring stability and effective task learning. The electron microscopy (EM) connectivity (of the V1 cortical column model) is obtained from Allen Institute (2023), which includes dendritic tree reconstructions and local axonal projections for hundreds of thousands of neurons, detailing their 0.5 billion synaptic connections. From this, we selected 198 cells, focusing on fully proofread neurons closest to the midpoint between layers 2/3 and 4. Connectivity strength for each neuron is determined by summing the volume of each post-synaptic density to target cells, distinguishing between excitatory and inhibitory cell types. For instance, if cell 'a' forms 10 synapses with cell 'b', the connection strength of connection[a,b] represents the combined volume of synaptic densities at cell 'b'. Inhibitory connections are assigned a sign of -1, while excitatory ones receive +1. For the Dale's law obeying initial connectivity, balanced initialization was done following the process in Yang & Wang (2020) with 80% excitatory and 20% inhibitory neurons (see the notebook $EIRNN.ipynb$).

It is crucial to highlight that, in testing our Theorem, which examines the effect of the **initial** weight rank, all low-rank modifications are not enforced during training (although the impact of enforcing these structures could be an interesting avenue for future exploration). Weights are adjusted freely based on gradient descent learning.

### C.2  TASK AND TRAINING DETAILS

Our code is accessible at `https://github.com/Helena-Yuhan-Liu/BioRNN_RichLazy`. We used PyTorch Version 1.10.2 (Paszke et al., 2019). Simulations were executed on a computer server with x2 20-core Intel(R) Xeon(R) CPU E5-2698 v4 at 2.20GHz, with the average task training duration being around 10 minutes. Following the procedure in George et al. (2022), which delved deeply into effective laziness metrics, we employed gradient-descent learning with the SGD optimizer. Unless mentioned otherwise, the learning rate was $3e - 3$, but we validated that our findings remain consistent across various learning rates (see Appendix D). For stopping, we trained the neurogym tasks for 10000 SGD iterations, which led to comparable terminal losses and accuracies across initializations. For the sMNIST task, we concluded our training upon reaching 97% accuracy, a criterion informed by both published results and our computational resources. We also experimented with halving and doubling the training iterations and observed similar trends. All weights — input, recurrent and readout — were trained. For statistical analysis and significance tests, we used methods in the SciPy Package (Virtanen et al., 2020).

For the neuroscience tasks, we adopted the Neurogym framework (Molano-Mazon et al., 2022). Within this paper, these tasks are denoted as "2AF", "DMS", and "CXT", mirroring Neurogym

settings: $task =' PerceptualDecisionMaking - v0'$, $task =' DelayMatchSample - v0'$, and $task =' ContextDecisionMaking - v0'$, respectively. To expedite simulations and facilitate numerous runs, we operated with $dt = \tau_m = 100ms$ and abbreviated task durations: for 2AF, settings were $stimulus = 700ms$ and $decision = 100ms$; for DMS, they were $sample = 100ms$, $delay = 500ms$, $test = 100ms$, and $decision = 100ms$; for CXT, they comprised $stimulus = 200ms$, $delay = 500ms$, and $decision = 100ms$. For these three tasks, we used a batch size of 32 and trained for 10000 iterations.

Regarding the sequential MNIST task LeCun (1998), we employed a row-by-row format to hasten simulations. Inputs were delivered via $N_{in} = 28$ units, each presenting a row's grey-scaled value, culminating in 28 steps with network predictions rendered at the final step. Training hinged on the cross-entropy loss function; targets were provided throughout training for the neuroscience tasks, as per Neurogym implementation, and targets were provided at only a trial's conclusion for the sequential MNIST task. For this task, we used a batch size of 200 and trained for 10000 iterations.

For the student-teacher two-layer linear network simulations in Figure 4A and Figure 3, we set $N = 1000$, $d = 2$ (also found similar trends for $d = 20$ and $d = 100$), $z = Fx$, all entries of $w$ (or $\beta$ when $F = I$) to 1 and entries of $X$ are sampled from a uniform distribution over the interval $[-2, 2]$. We used standard Normal initialization for both $W_1$ and $W_2$ with $\sigma = 0.001$. For Figure 3, $F$ is constructed from SVD, i.e. $F = USV^T$, with $U$ and $V$ generated from arbitrary orthogonal matrices, and $S$ is a diagonal matrix consisting of the singular values with the top half of the singular values set to $\kappa$ and bottom half set to 1, where $\kappa$ is the condition number of $F$. For the aligned initialization, $W_1$ is initialized as given in Proposition 1 with $\beta = w^T F$ ($w$ here is illustrated in Figure 3), and the $F$ is replaced by its rank-$(d/2)$ truncation for the partially aligned initialization case. For the MNIST task shown in Figure 4B, we used a two-layer feedforward network with a ReLU activation function. The architecture consists of an input layer with 784 units corresponding to the image pixels, a hidden layer with 300 units, and a linear readout layer with 10 output units. The weights of the hidden layer and the readout layer were initialized similarly to the input and readout weights, respectively, used in the RNN settings.

# D   ADDITIONAL SIMULATIONS

We perform additional simulations to show the robustness of our main trends, Low-rank initial recurrent weights lead to greater changes (or effectively richer learning) in RNNs. We show the main trends observed in Figure 1 holds also for Uniform initialization (Figure 7), soft initial weight rank (Figure 8), various network sizes (Figure 9), learning rates (Figure 10), initial weight gains (Figure 11) and Dale's Law constraint throughout training (Figure 17), finer simulation time step $dt$ (Figure 12) and fixing the leading initial weight eigenvalue (Figure 13). The trends in Figure 2 also applies to the DMS task (Figure 5) and the CXT task (Figure 6). Also, without the low-rankedness in the shuffled EM connectivity, the impact on effective laziness also goes away (Figure 19). In Figure 4 we confirm that the results, shown in Figure 1 and predicted by Theorem 1, are also observed in a two-layer linear network setup. Again, we find that in situations where initializations are random, higher rank initialization leads to greater tangent kernel alignment than lower rank cases. We have also tracked the evolution of kernel task alignment (Figure 14) and kernel effective rank over the course of training (Figure 15).

**A   Student-teacher setup for two layer linear net**

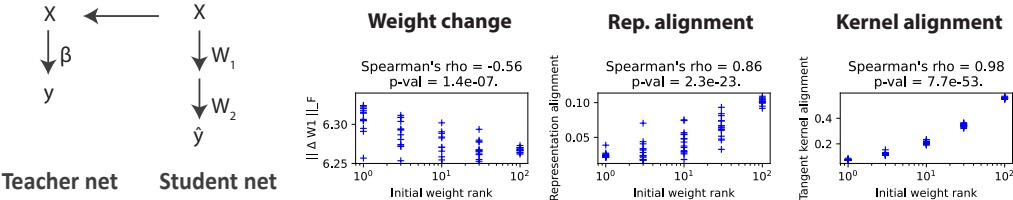

**B   Feedforward network in non-idealized setting**

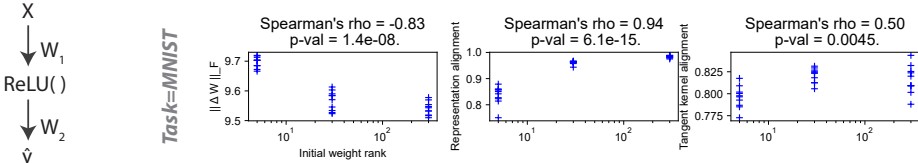

Figure 4: **As predicted by the theoretical results, higher rank random initialization leads to effectively lazier learning in two-layer linear network.** A) We use the student-teacher two-layer linear network setup described in Section 2.3. B) a non-idealized setting: two-layer feedforward network with ReLU activation and 300 hidden units trained on the MNIST dataset. Plotting convention follows that of Figure 1.

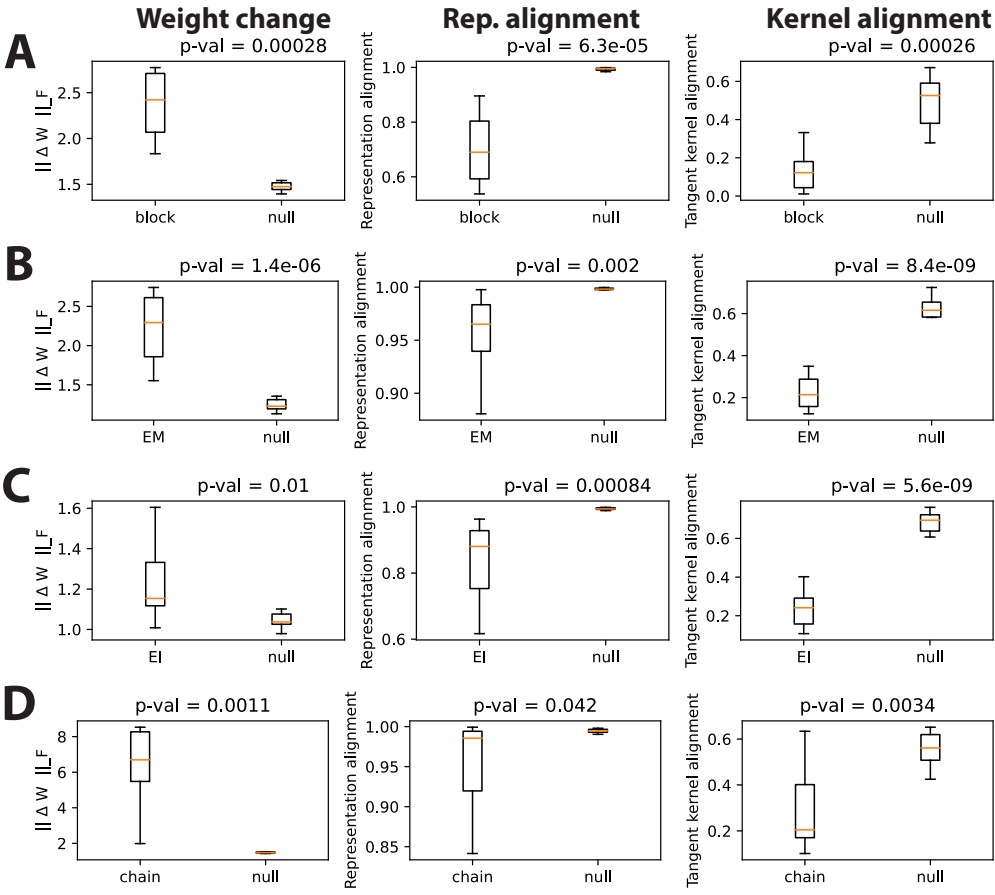

Figure 5: We repeated Figure 2 for the DMS task and observed similar trends: low-rank initialization, achieved by experimentally-driven initial connectivity in Figure 2, leads to effectively richer learning. The plotting conventions used here follow those in Figure 2, with panels A-D corresponding to the ones in that figure.

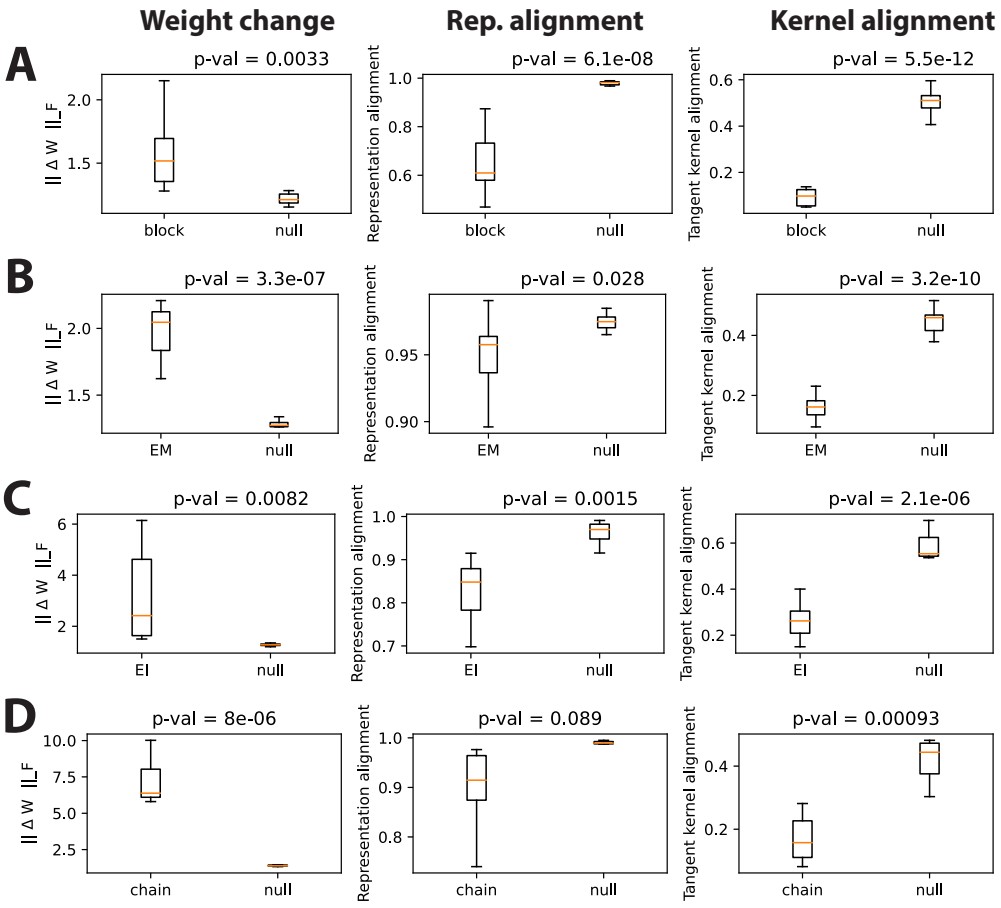

Figure 6: We repeated Figure 2 for the CXT task and observed similar trends: low-rank initialization, achieved by experimentally-driven initial connectivity in Figure 2, leads to effectively richer learning. The plotting conventions used here follow those in Figure 2, with panels A-D corresponding to the ones in that figure.

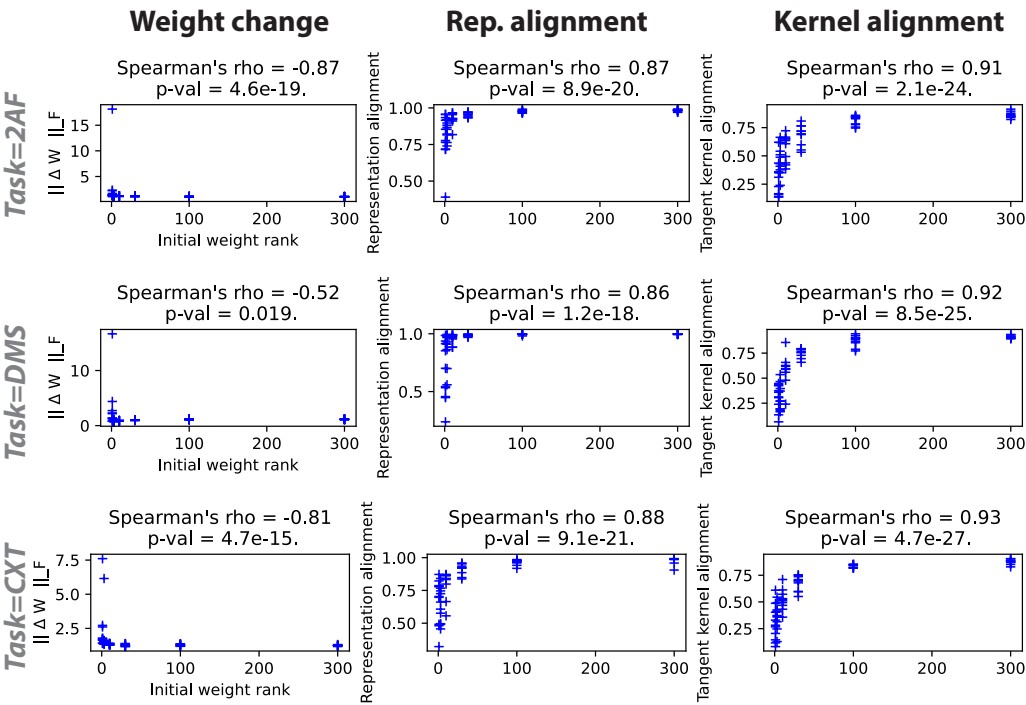

Figure 7: **Consistent trends observed in Figure 1 also for Uniform initialization**. We replicated the results of Figure 1 — where the initial weights follow a zero-mean Gaussian distribution $W_{ij} \sim \mathcal{N}(0, g^2/N)$ — but now for Uniform initialization $W_{ij} \sim \mathcal{U}\left(-\frac{g}{\sqrt{N}}, \frac{g}{\sqrt{N}}\right)$. Plotting conventions follow that of Figure 1.

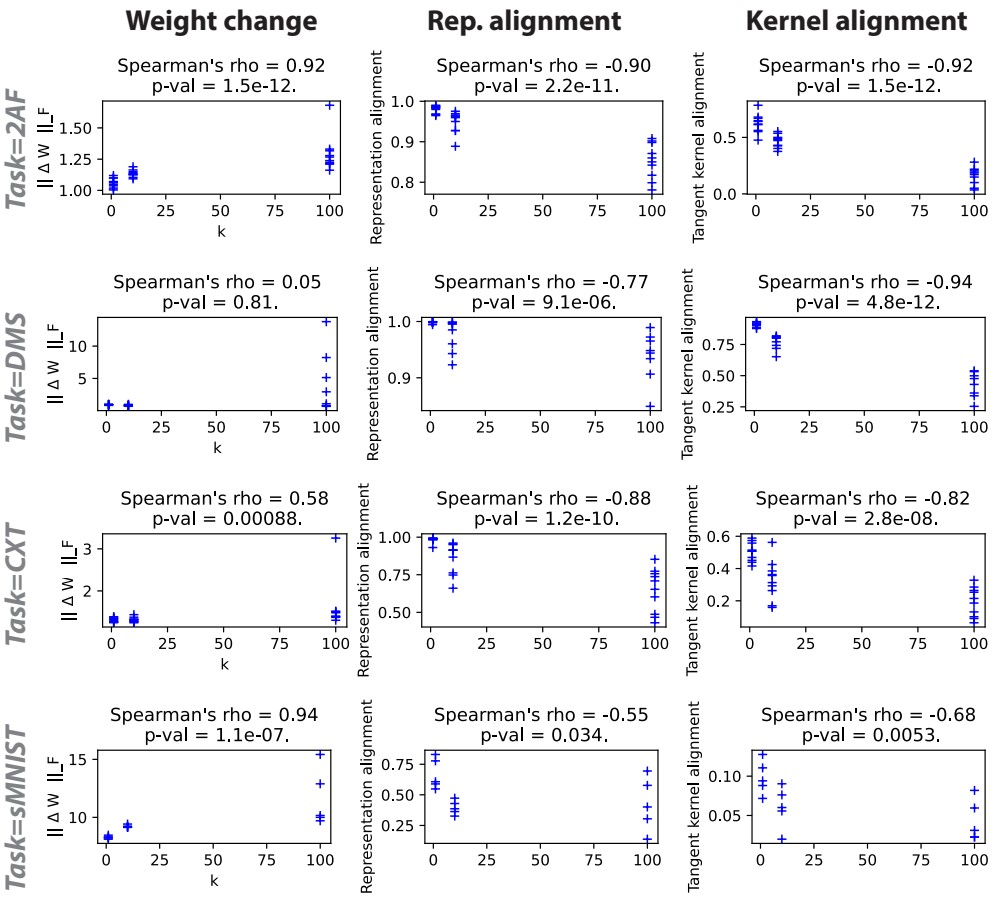

Figure 8: **Consistent trends observed in Figure 1 also for "softer" low-rank weights**. Here, instead of the "hard" low-rank weights in Figure 1 — where the $i^{th}$ weight singular value $s_i$ is set to $0$ if $i > r$ for rank $r$ — we introduce a smoother decay in singular value, where we replace the singular values with $s_i = s_1(1 - i/N)^k$ after performing SVD; this means that greater $k$ leads to lower effective rank. Plotting conventions follow that of Figure 1

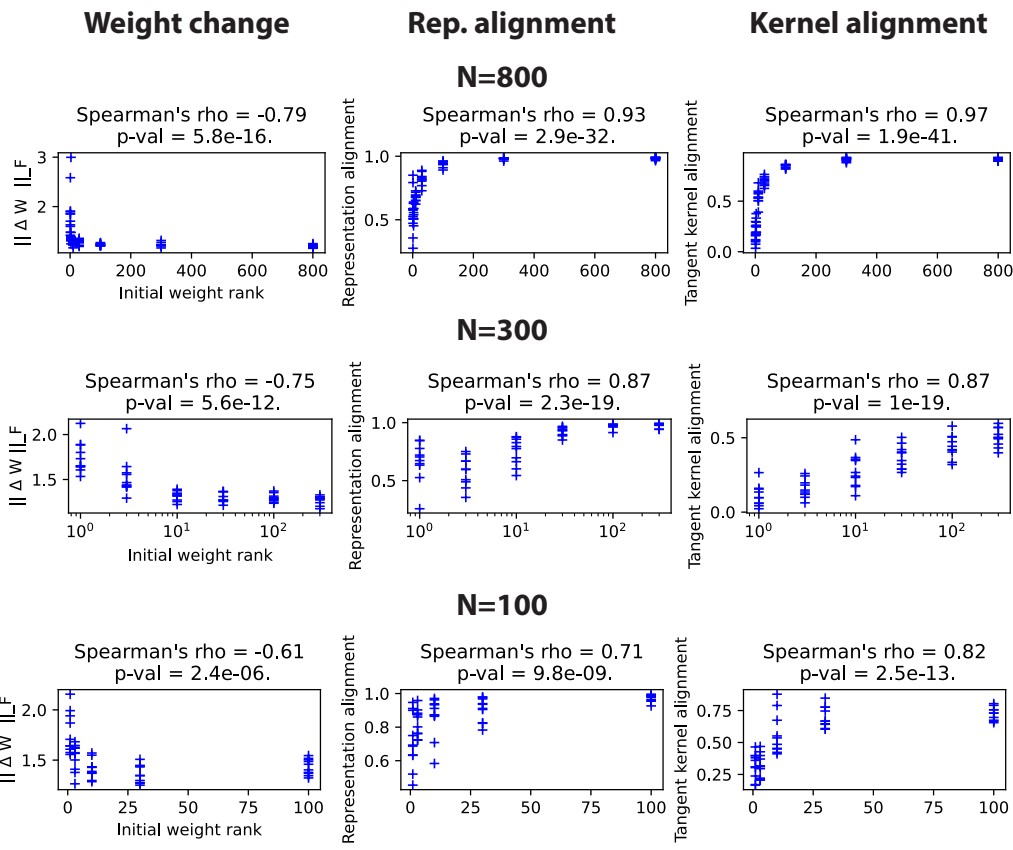

Figure 9: **Consistent trends observed in Figure 1 across various network sizes** ($N$). We replicated the results of Figure 1 for different values of $N$, using the CXT task as an illustrative example. However, the observed trend remains consistent for both the 2AF and DMS tasks. Plotting conventions follow that of Figure 1.

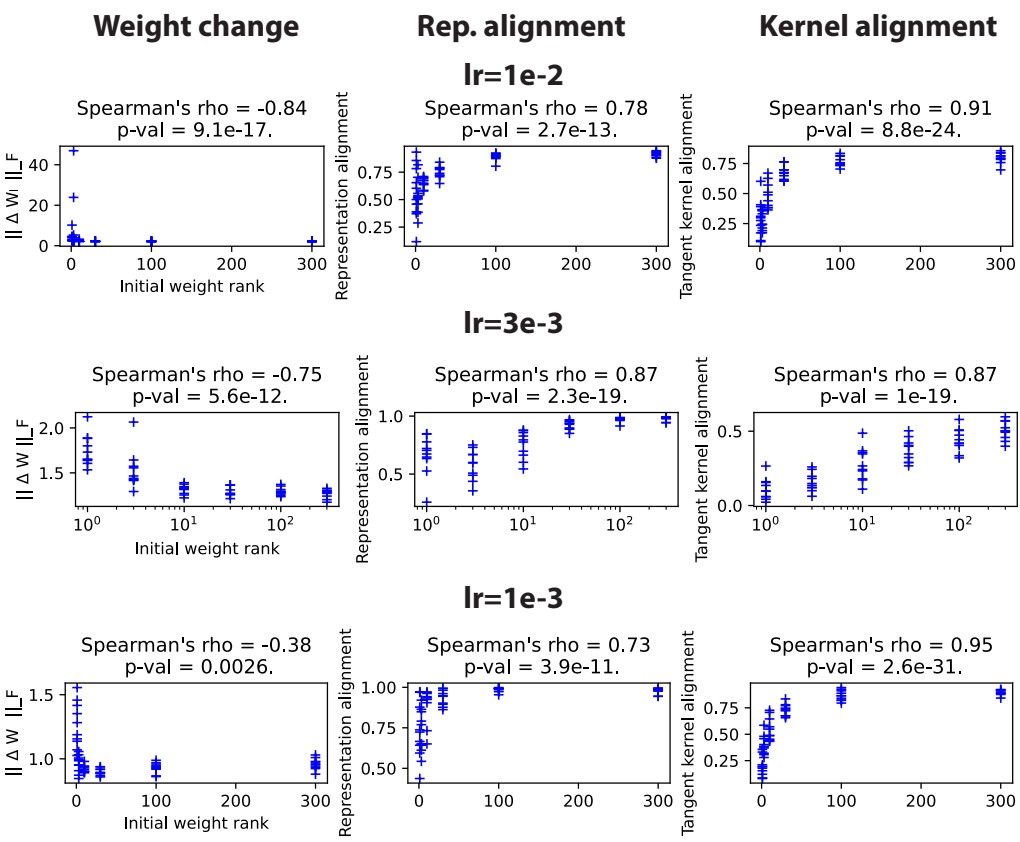

Figure 10: **Consistent trends observed in Figure 1 across various learning rates (lr).** We replicated the results of Figure 1 for different learning rates, using the CXT task as an illustrative example. However, the observed trend remains consistent for both the 2AF and DMS tasks. Plotting conventions follow that of Figure 1.

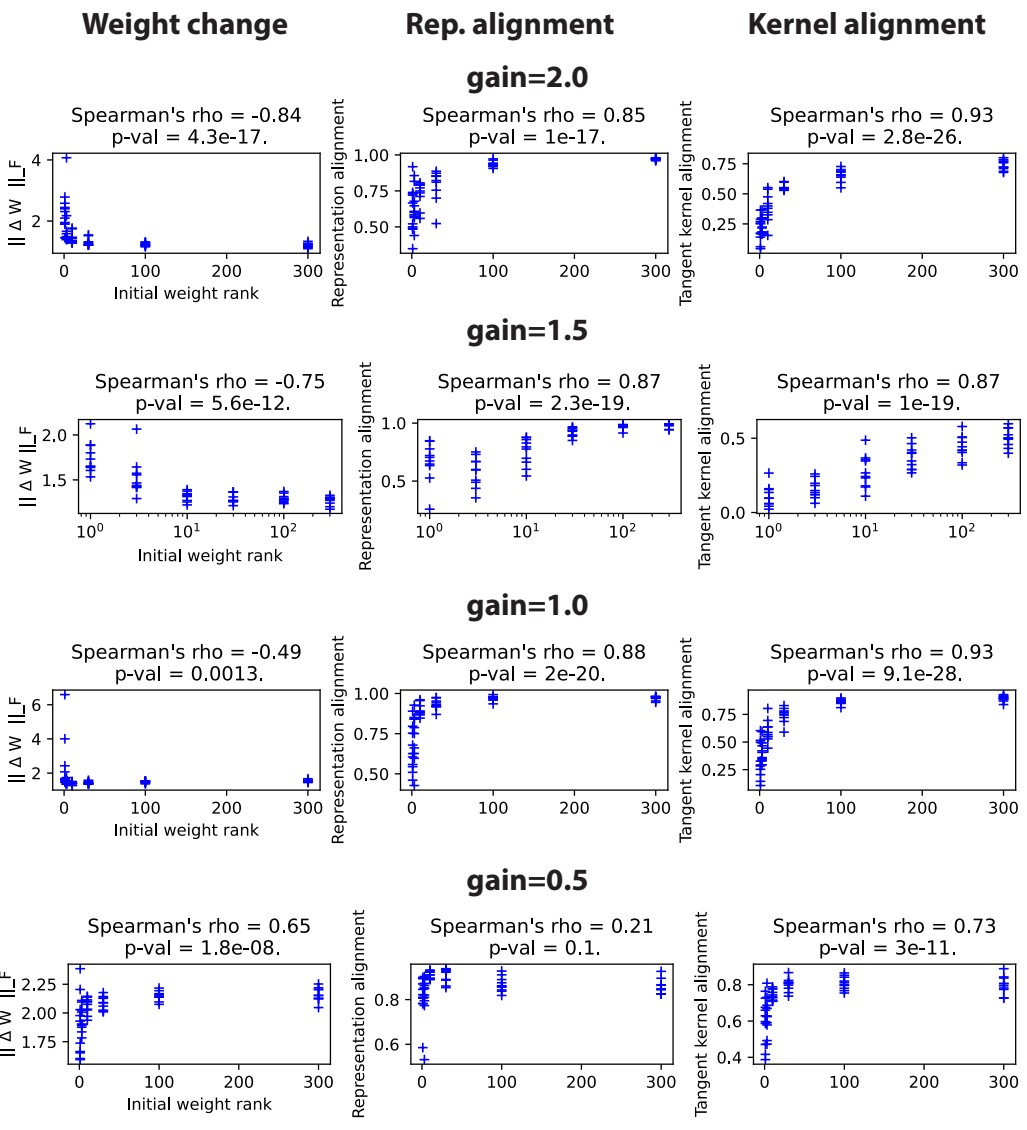

Figure 11: **Consistent trends observed in Figure 1 across various initial gain**. Here, the gain refers to $g$, as weights are initialized as $W_{ij} \sim \mathcal{N}(0, g^2/N)$. The trends hold for most typical range of $g$ from 1.0 to 2.0, but gets weakened for smaller values, $g < 1.0$ (a closer examination of the regime bias in such setting in RNNs is left for future work). We replicated the results of Figure 1 for different learning rates, using the CXT task as an illustrative example. However, the observed trend remains consistent for both the 2AF and DMS tasks. Plotting conventions follow that of Figure 1.

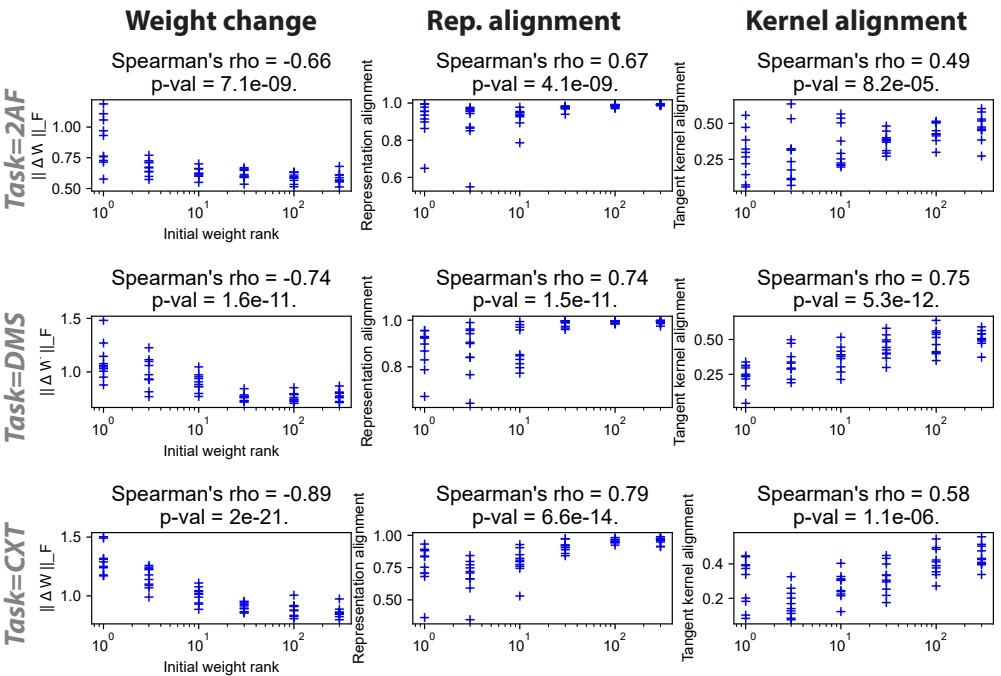

Figure 12: **Trends in Figure 1 are also observed In training RNNs with a fivefold finer time step ($dt$) and a sequence length extended by five times**. As expected, higher rank initializations led to a marked increase in effective laziness. Plotting conventions follows that of Figure 1.

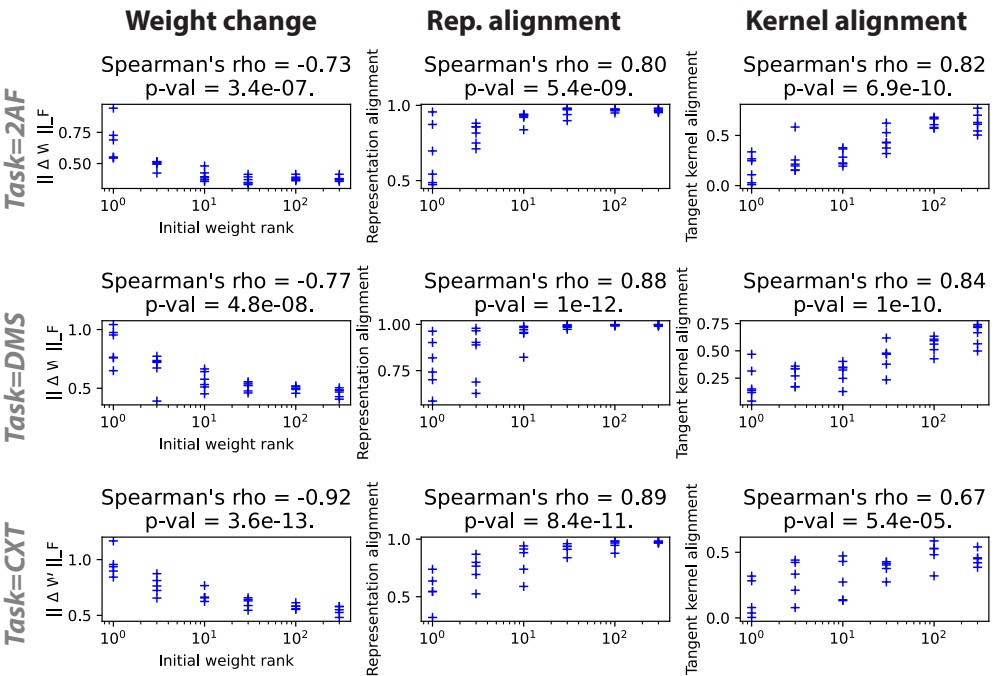

Figure 13: **Trends in Figure 1 are also observed when fixing the leading weight eigenvalue instead of the Frobenius norm across comparisons**. As expected, higher rank initializations lead to effectively lazier learning. Plotting conventions follows that of Figure 1.

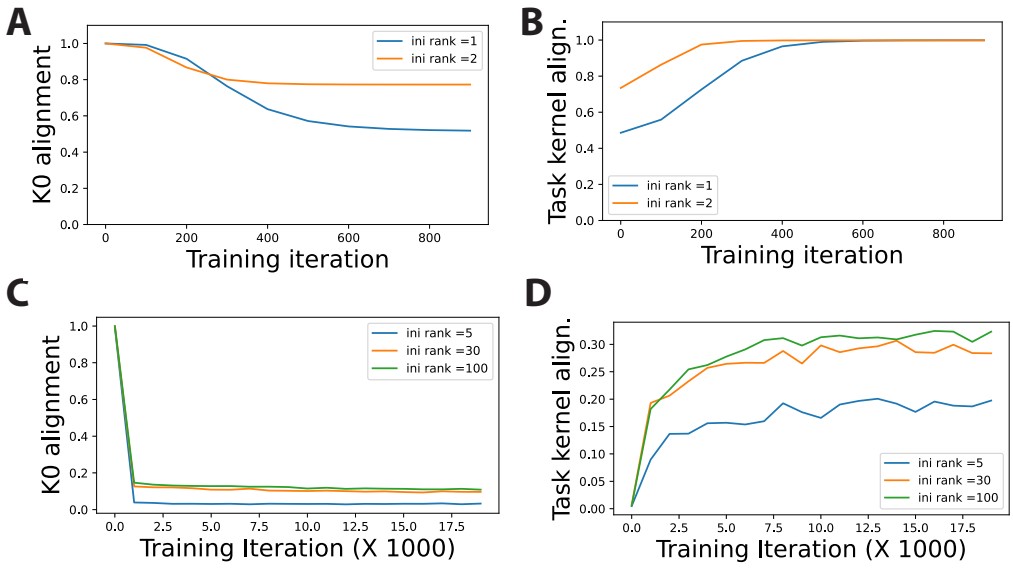

Figure 14: **[A-B] The idealized two-layer linear network setting from Fig. 2 in Atanasov et al. (2021).** A) Examining the K0 alignment — the alignment between the kernel at various training iterations and the initial kernel — reveals that low-rank random initialization leads to greater changes during training; here, different curves correspond to different initial weight ranks. B) Despite these greater changes, networks with low-rank random initialization take longer to align with the task, as shown by the task kernel alignment metric $y^T K y / |y|^2 Tr K$ throughout training. We remind the reader that $y$ corresponds to the target output and $K$ corresponds to the NTK. **[C-D] A non-idealized setting: the sMNIST task.** C) This panel shows similar trends to A). D) Similar to B), lower-rank random initializations do not achieve as high task kernel alignment within the trained iterations. This is measured by the centered kernel alignment (CKA), which assesses the kernel's alignment with class labels (Eq. 7 in Baratin et al. (2021)). Although higher CKA values during training could suggest enhanced feature learning (characteristic of the standard rich regime), this aligns with our findings on the effective learning regime, which focuses on changes post-training (see Introduction). Our theory in Section 2.3 suggests that lower-rank initializations require greater changes to align with the task, which would typically require more training iterations, as seen in panel B. If training is halted prematurely, perhaps due to resource constraints (as in panel D), these initializations may achieve lower final alignment within the training period. It remains unclear if extended training would lead to similar final alignment across different initializations in a wide range of scenarios. Future research should further investigate the relationship between rankedness of initializations and their impact on the converged solution's representation, including task kernel alignment, across diverse settings.

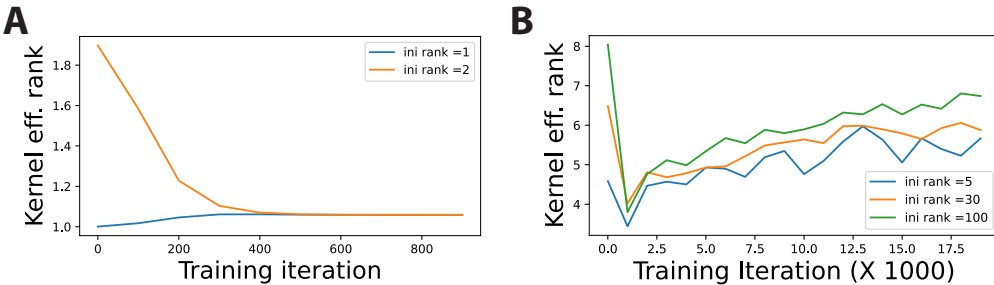

Figure 15: The evolution of the leading NTK eigenvalue relative to the rest of the eigenvalues was tracked using an effective rank measure. This measure is based on the ratio of the kernel trace to the kernel dominant eigenvalue, i.e., $\sum_i \lambda_i / \lambda_1$, which indicates the number of eigenvalues on the order of the dominant one. We apply this analysis to A) the idealized setting and B) the sMNIST task, as used in Appendix Figure 14 and Figure 1, respectively. These results suggest that the kernel effective rank approaches that of the task throughout the training process.

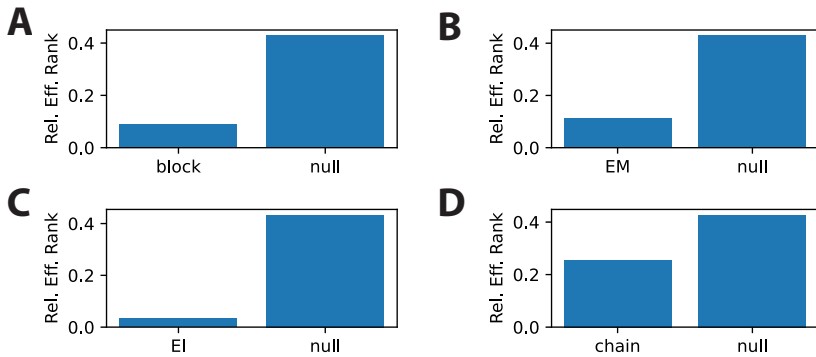

Figure 16: Measuring connectivity effective rank based on singular values instead of eigenvalues led to a similar conclusion as Figure 2: these experimentally-driven connectivity structures exhibit lower effective rank compared to random Gaussian initialization (null). The plotting conventions used here follow those in Figure 2, with panels A-D corresponding to the ones in that figure.

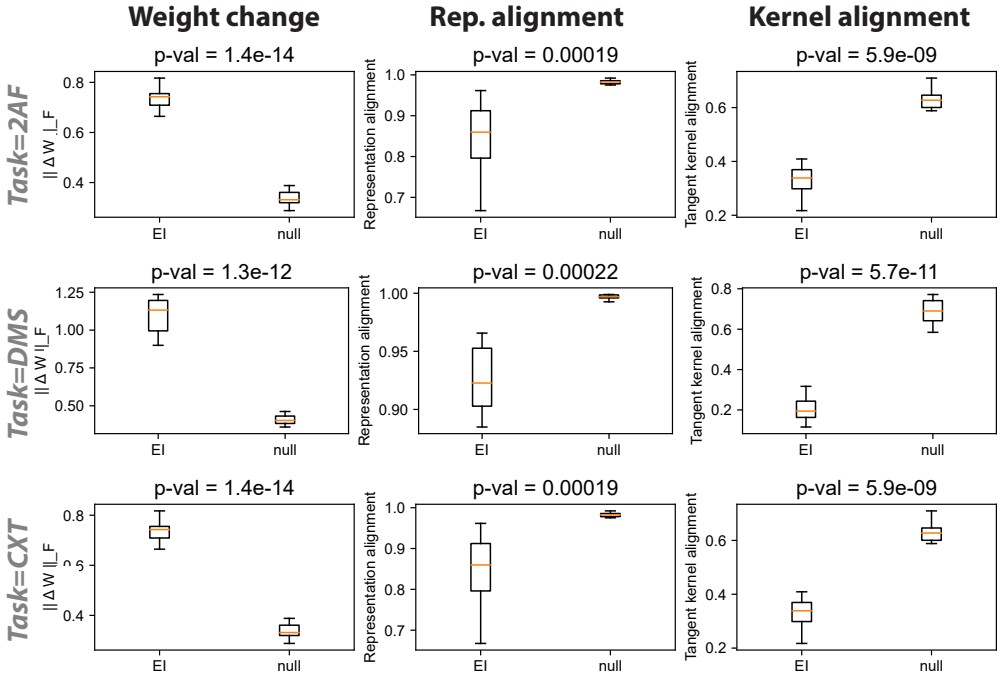

Figure 17: Maintaining the constraint of Dale's Law during the entire training process, rather than just at initialization, produced a trend analogous to that observed in Figure 2. Plotting conventions follow that of Figure 2.

Figure 18: Training RNNs on the pattern generation task, as illustrated in Fig. S7 of Bellec et al. (2020), showed consistent trends with our conclusion: initializations with higher ranks resulted in a more pronounced tendency towards effectively lazier learning. Plotting conventions follows that of Figure 1.

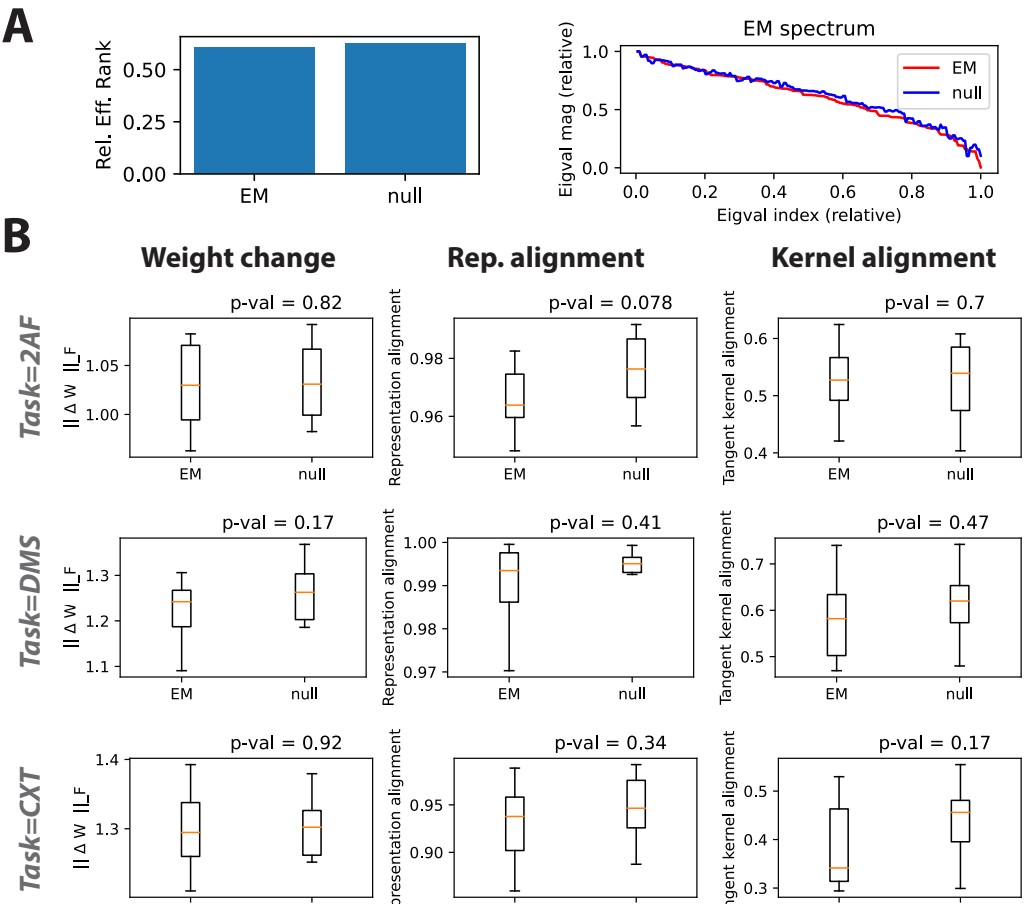

Figure 19: **Shuffling the EM connectivity, while maintaining the sparsity structure, destroys the low-rankedness and the impact on effective laziness**. We repeated the analyses with the EM initial connectivity in Figures 2 but performed random shuffling on the EM connectivity, to see if the low-rankedness and the impact on effective laziness is due to the sparsity in the dataset. Performing such shuffling destroys these trends. Plotting conventions follow that of Figure 2