# OpenReview forum: "How connectivity structure shapes rich and lazy learning in neural circuits"
_ICLR.cc/2024/Conference — ICLR 2024 poster_

### Official Review · Reviewer_rK41 · 2023-10-23

**Soundness:** 3 good
**Presentation:** 3 good
**Contribution:** 3 good
**Rating:** 8
**Confidence:** 4

**Summary:**

This paper probes the effect of weight initialization rank on feature learning in recurrent neural networks. The authors study in particular how the alignment between a particular low-rank initialization and the task affect how much the kernel moves during training.

**Strengths:**

In the large, I think this paper is timely, and the core idea is (to the best of my knowledge) novel and interesting.

**Weaknesses:**

1. Upon reaching the conclusion of the paper, I found myself confused as to why the authors did not perform any experiments with feedforward networks. Though I appreciate the neuroscience-inspired focus on recurrent networks, including some tests in the feedforward setting would be valuable. In particular, this would show more directly how departures from the idealized setting of their theoretical results (linear networks with whitened data) affect the phenomenology. Demonstrating that these ideas are applicable to deep feedforward networks  and more standard machine learning tasks would in my view substantially improve the impact of the paper by tying it more closely to the main body of theoretical work on feature learning.

2. The metrics introduced in Section 2.2 are restricted to measuring changes in weights or kernels over the course of training. Particularly given the fact that the stated goal is to investigate how task structure affects feature learning, it would be useful to also track a measure of task-kernel alignment, for instance the kernel-target alignment $y^{\top} K y / |y|^2 tr K$ or the task norm $y^{\top} K^{-1} y$ throughout training.

3. The criteria used for task selection are unclear. The authors choose three of the original tasks from the neurogym environment (I note that this task suite has been critiqued by Khona et al., "Winning the Lottery With Neural Connectivity Constraints: Faster Learning Across Cognitive Tasks With Spatially Constrained Sparse RNNs") along with sequential MNIST. Why are these four tasks relevant? The paper could be made more compelling either by using a more diverse suite of tasks or by justifying why these four are appropriate.

4. On the whole, I found the Discussion to be a weak point of the manuscript. Section 4.1 does not answer the vital question of precisely what experimentally-measurable biological phenomena the present work can explain, contextualize, or predict. As it stands, this section is in some sense a restatement of Zador's ideas around the importance of task-aligned initialization in the language of lazy learning. Moreover, the discussion of implications for deep learning promised by the section title seems to be missing. Section 4.2 contains far too many ideas to be jammed into its single paragraph; for clarity the authors should split it into at least two paragraphs. Finally, the relevance to neuroscience of neural collapse and of Tishby's proposals regarding the information bottleneck needs to be justified. The sentence where they are mentioned is much too long as it is, and these are both somewhat subtle and controversial topics.

**Questions:**

1. In the block of citations in the second sentence of the introduction, the work of Canatar, Bordelon, and Pehlevan (2021) should be cited before Xie et al 2022, as the latter is based on the results developed in the former work.

2. Is "tabula rasa" the appropriate way to describe Gaussian or Uniform random initialization with non-vanishing variance?

3. In-text citations to the work of the Allen Institute and the MICrONS Consortium are not formatted correctly.

4. Why do the eigenvalues in Figure 2 appear not to be sorted by magnitude? Also, why is $\frac{\sum\_{i} |\lambda|\_{i}}{|\lambda\_{1} N}$ the relevant notion of effective rank in this setting?

5. Figures 2 and 3 each occupy more than half of a page, but both contain a significant amount of whitespace. Is it possible to combine them side-by-side into a single figure?

---

> ### Author Response · Authors · 2023-11-16
> **Response to Reviewer rK41 (1/3)**
>
> We thank the reviewer for the accurate summary of our work and recognizing its timeliness and novelty. Moreover, we thank the reviewer for their concrete and executable suggestions to sharpen our claims and improve the presentation. We are confident that by incorporating the reviewer’s insightful feedback, our updated manuscript has been significantly improved along all axes of soundness, presentation and contribution.
>
> **Experimentation with feedforward networks in a non-idealized setting**: as the reviewer pointed out, we studied RNNs due to our neuroscience focus, and we completely agree with the reviewer that adding results on feedforward settings would be important. We agree with the reviewer that this would help out with tying to the existing literature on feature learning predominantly in feedforward settings. Following the reviewer’s suggestion to showcase this in non-idealized feedforward settings, we have now appended a panel to a figure in Appendix (see Appendix Figure 9B), showing our main claim (higher rank initialize lead to effectively lazier learning) also apply to to nonlinear feedforward networks trained on a real-world dataset. Although we trained the network on MNIST (instead of something like CIFAR), it is sufficient for the purpose of departing from the idealized setting. As originally mentioned in Limitations & Future Work, more comprehensive examination across wider range of architecture, including feedforward settings, is left for future work. We have also added a sentence in Simulation Results alerting interested readers to this appendix figure.
>
> **Tracking task-kernel alignment during training**: We thank the reviewer for the insightful comment. In line with the reviewer’s suggestion of tracking impact on not just the final trained network but also during the learning process (and related comments by other reviewers), we have now added Appendix Figure 14 tracking the tangent kernel alignment as well as the alignment of the kernel and task throughout training. Figure 14A examines the kernel alignment (alignment of the current kernel to the initial one) in the idealized linear setting with 2D input in Fig. 2 in Atanasov et al. across different initialization schemes: random rank-1 initialization, random full-rank initialization and initially aligned low-rank initialization. We see that low-rank random initialization lead to greater movements, i.e. growing to be more dissimilar to the initial kernel (lower alignment value). Figure 14B looks at the task kernel alignment, $y^T K y / |y|^2 TrK$. Regardless of the initializations, the network kernel moves to be more aligned to the task over training. Low-rank initialization, without being specifically tuned to the task, is the latest to achieve alignment.
>
> Figure 14C examines the kernel alignment (alignment of the current kernel to the initial one) in a non-idealized setting: sMNIST task. Again, random low-rank initialization leads to more movement over training, i.e. more dissimilar to the initial kernel. For Figure 14D, with multiple (10) output targets, we track the centered kernel alignment (CKA) of the NTK and class label, e.g. used in Baratin et al. (2021). We again saw that all initializations grow to be more aligned with the class label through training, but higher rank initialization gets to a higher alignment value faster and with less kernel rotation. We have already trained for nearly 20000 SGD iterations and we are not sure if training longer would eventually lead to similar final alignment values across initializations. We note that although achieving higher CKA over training can indicate greater degree of feature learning, a hallmark of the standard rich regime, this does not contradict our findings centered on effective learning regime that is defined based on the amount of change post-training (see Introduction). To incorporate these new findings in the main text, we have added sentences in Simulation Results to bring these Appendix Figures to the attention of an interested reader.
>
> **Please continue to the comment below**

---

> ### Author Response · Authors · 2023-11-16
> **Response to Reviewer rK41 (2/3)**
>
> **Improving Discussion Section**: In response to the reviewer’s crucial comment regarding more concrete experimental explanation or prediction, we have now updated the manuscript, and the paragraph reads as: “*We study the impact of effective weight rank on learning regime due to its potential implications in neuroscience. In particular, learning regimes are indicative of the amount of change undergone through learning, which bear consequences for metabolic costs and catastrophic forgetting (McCloskey & Cohen, 1989; Plaçais & Preat, 2013; Mery & Kawecki, 2005). These considerations are especially pertinent when we examine the various learning regimes evident in neural systems. For instance, during developmental phases, neural systems undergo resource-intensive, plasticity-driven transformations characterized by significant synaptic changes. These transformations are in contrast to the more subdued adjustments observed in mature neural circuits (Lohmann & Kessels, 2014). Based on this understanding, we predict that a circuit's alignment with specific tasks is likely established either through evolutionary processes or during these early developmental phases. As such, the specialization of a neural circuit (e.g., ventral vs dorsal (Bakhtiari et al., 2021)) likely stems from its engagement with tasks that share overlapping computational requirements, ensuring that each circuit is optimally configured for the tasks it is predisposed to perform. Conversely, circuits with high-rank structures, due to their inherent flexibility, are expected to be less specialized, potentially engaging in the learning of a broader range of tasks. With this in mind, our framework could be used as a tool to compare the connectivities across brain regions (and species) to predict their function and flexibility…* ”
>
> For deep learning, we note that while low-rank initialization is not common practice, low-rank adaptation and other update parametrizations have recently been popular in large model training schemes. See for example, Low-rank Adaptation (LoRA) in Hu et al. (2021). As such, we argue that the impact of low-rank structures on learning dynamics is quite relevant for current AI practices, and our study contributes to examining how having low-rank structures impact the effective learning regime. The methods and results we present here have the potential to be adapted and built upon to study rankedness and learning regime in several settings, and we are keen to see future work stemming from these ideas.
> We also note that our study has the tangential benefit of informing the deep learning community about potential applications of their models in other fields such as neuroscience. However, due to the significantly heavier balance of Section 4.1 in discussing neuroscience implication, we have removed “Deep Learning” in the section heading. We have updated Section 4.1 reflecting these points.
>
> Upon re-reading Section 4.2, we agree with the reviewer that Section 4.2 is jammed with too many ideas. As per the reviewer’s suggestion, we have now broken 4.2 into smaller paragraphs for readability. We also thank the reviewer for pointing out the long sentence involving neuroscience implications and Tishby’s proposal. This sentence suffers the same problem as the entire section that the reviewer pointed out: too many ideas jammed together, which compromises clarity. We, thus, have removed the mention of Tishby’s proposal since it is creating confusion and distraction from our main point in the sentence, which is asking for deeper exploration into the neuroscience implications in the future.
>
> **Please continue to the comment below**

---

> ### Author Response · Authors · 2023-11-16
> **Response to Reviewer rK41 (3/3)**
>
> **Unclear task selection criteria**: We thank  the reviewer for this valuable comment. The primary criteria for our task selection were neuroscience relevance, aligning with our core objectives and benchmark popularity, to ensure we weren't perceived as choosing idiosyncratic tasks that merely optimize our results — a rationale that also informed our decision to include sMNIST. That said, there are many tasks that fit the criteria due to the wide range of tasks that different species can solve, and the type of tasks we examine fall far short of the pool of all relevant tasks, so we listed the examination of wider range of tasks and architecture as limitation and future work in our initial submission. We remark that because of Theorem 1 and our intuition explained at the beginning of Discussion, we anticipate that our conclusion should hold across a wide range of settings despite having not exhaustively explored all relevant settings. In response to the reviewer’s suggestion, we have revised our Discussion section to reflect this and make propositions for related future work.
>
> Despite that, the reviewer’s feedback made us realize that the strength of our empirical results would be improved if we can show at least one additional task (with neuroscience relevance) that is significantly different from the structure of the existing tasks. For that reason, we have added the sequence generation task inspired from Bellec et al. (2020) (Appendix Figure 15) and a sentence in Discussion alerting interested readers to that figure. We remark that exploration across a broader range of tasks is left for future work, as mentioned in our initial submission, and in particular, it would be interesting to explore the harder neuroscience tasks in Mod-Cog introduced by Khona et al. in the future (we have added the citation of Mod-Cog to that discussion in our updated Limitations & Future Directions).
>
> **Citing Canatar et al.**: We thank the reviewer for catching this missed citation. We worked hard to be comprehensive in our citation, and yet, we still somehow let this highly relevant reference slip through our attention. We have thus added it to exactly where the reviewer suggested.
>
> **Tabula rasa**: we thank the reviewer for bringing this to our attention. "Tabula rasa" means "blank slate”. However, Gaussian or Uniform random initialization, with non-vanishing variance, is not exactly a “blank state” in the strictest sense, as they can still exhibit inherent bias. We have thus replaced “tabula rasa” with “random” in the offending sentence.
>
> **Citations to Allen and MiCrONS**: we thank the reviewer for catching these and we have now fixed them.
>
> **Sorting eigenspectrum plots and effective rank**: We thank the reviewer for the comment. The eigenvalues were not sorted because we used *numpy.linalg.eigvals()*, which does not guarantee sorted eigenvalues and we didn’t ensure that once we were able to clearly observe the faster decay trend in the eigenspectrum of the experimentally-driven connectivities was still visible. As per the reviewer’s comment, we have now sorted the eigenvalues in the eigenspectrum plots.
>
> As for the effective rank measure — which captures the proportion of eigenvalues on the order of the leading one — we chose it because it has been used before (e.g. Murray et al., ICLR’23) and would correspond to the area under the curve of the eigenspectrum plots scaled relative to the leading one, thereby informing us regarding the eigenspectrum decay. Low rank matrices would have fewer eigenvalues on the order of the leading one. That said, singular values are used more generally for capturing the effective rank. Hence, we have also added plots that measure effective rank using singular values (Appendix Figure 13) and a sentence in Simulation Results referring to that figure. These plots all support our main point here: these example experimentally-driven structures exhibit lower effective rank compared to null.
>
> **Combining Figure 2 and 3**: We thank the reviewer for this suggestion and we have now combined them into one figure. This indeed gave us a lot more space to address the reviewers’ important points.

---

> > ### Comment · Reviewer_rK41 · 2023-11-16
> > **Response to author response**
> >
> > I thank the authors for their thoughtful and detailed reply to my comments and those of the other reviewers. I think the paper is substantively improved by these changes, and therefore will raise my score.
> >
> > One small comment - the heading of Section 4.1 in the currently-visible manuscript reads "POTENTIAL IMPLICATIONS TO BOTH NEUROSCIENCE;" the "both" should be removed along with the "deep learning" that you already cut.

---

> > > ### Author Response · Authors · 2023-11-16
> > > **Thank you**
> > >
> > > Thank you again for your constructive and insightful feedback. We are especially appreciative of your responsiveness and consideration in revising your score after reading our response. We have also removed "both" from the Section 4.1 heading.

---

> > > > ### Author Response · Authors · 2023-11-16
> > > > **Further possible improvements to further improve score?**
> > > >
> > > > We are grateful for the reviewer's recognition of the improvements we brought to the revised manuscript. We addressed the major points raised by the reviewer and kindly ask if there are other points we could address for the reviewer to consider increasing their score further beyond the borderline acceptance region.

---

> > > > > ### Comment · Reviewer_rK41 · 2023-11-16
> > > > > **Thanks for the note, score raised**
> > > > >
> > > > > I thank the authors for their note. I think the paper should be accepted as is, so have raised my score to 8.

---

### Official Review · Reviewer_SwWy · 2023-10-31

**Soundness:** 3 good
**Presentation:** 3 good
**Contribution:** 2 fair
**Rating:** 6
**Confidence:** 4

**Summary:**

The work investigates how the initial weight structure, especially its (effective) rank, influences network learning dynamics, and in particular whether the network learns in the "lazy" (small change in tangent kernel) or "rich" (substantial evolution of tangent kernel) regimes. The paper is written using neuroscience as a motivation, citing the fact that connectivity in the brain is substantially more structured and lower-rank than standard initializations used in ML, and thus existing work on lazy/rich regimes may not apply directly to biological learning. The authors find that low-rank initializations typically lead to richer learning, unless the low-rank initialization happens to align with the structure of the task.

**Strengths:**

The paper is clearly written and technically sound.  The experiments provide solid evidence for the main takeaway of the paper (that lower-rank initializations generally lead to richer learning dynamics), using a variety of tasks and complementary measures of laziness/richness.  Figure 4 provides helpful intuition for understanding this conclusion, suggesting that low-rank initializations tend to incentivize richer learning because there is a greater need for it when the model's kernel at initialization has low alignment with the task structure.

**Weaknesses:**

I have two main concerns about this paper:

1. Given that the proof of Theorem 1 is left to an Appendix, the paper would benefit from providing more explanation / intuition for the main result.  I can imagine an intuition that the network "needs to" adjust its representation more to address the misalignment.  But a more thorough presentation of the key proof steps in the main text, and/or illustrative examples to provide intuition for the learning dynamics, would be very helpful.

2. It is not clear to me how the findings of this paper will be useful to either the ML or neuroscience communities.  I do not mean to claim that they aren't useful, but rather that the paper does not present a clear case for why they are.  From an ML standpoint, low-rank initializations are uncommon.  From a neuroscience standpoint, the applicability of this theoretical framework seems uncertain given that the learning algorithms used in the brain are not well characterized and may differ substantially from SGD.  Moreover, it is not clear to me what we gain by knowing that low-rank neural connectivity gives rise to richer learning than a counterfactual in which brains had higher-rank connectivity.  Is the idea to compare between brain regions with connectivities of different ranks to make predictions about different representation learning dynamics?  Or across species?  More concrete and precise proposals for how these findings could be used would be helpful.

3. The richness/laziness of learning dynamics by itself tells us very little about the representations a network learns.  If the goal is to understand the impact of weight initialization on representation learning, the paper would benefit from more direct consideration of this question.  For instance, see the question below.

**Questions:**

Do the kernels associated with low-rank-initialized networks grow more aligned with the target function than those of high-rank-initialized networks?  Or do they merely grow "as aligned" as the lazier networks with high-rank initializations?

---

> ### Author Response · Authors · 2023-11-16
> **Response to Reviewer SwWy (1/2)**
>
> We thank the reviewer for their positive feedback on the presentation and soundness. More importantly, we thank the reviewer for their concrete suggestions to further improve upon these axes as well as  to better articulate our contribution. Below, we explain how we address the reviewer’s concerns.
>
> **More explanation/intuition of the main theoretical result**: we thank the reviewer for this important comment. Our intuition is indeed in line with that of the reviewer’s. As per the reviewer’s suggestion, we have added a sentence explaining the intuition right after the Theorem statement. The sentence reads as “*The intuition of Theorem 1 result is that, when two random vectors are drawn in high-dimensional spaces, corresponding to the low-rank initial network and the task, the probability of them being nearly orthogonal is very high; this then necessitates greater movement to eventually learn the task direction*“. We have also updated the proof in the Appendix with more explanations for clarity.
>
> In addition, we followed the reviewer’s suggestion and updated the main text with an outline of key proof steps; roughly these steps are: 1) The kernel alignment (Eq. 9) consists of three factors: $Tr(K^{(0)} K^{(f)})$, $\| K^{(0)} \|$ and $\| K^{(f)} \|$. We first derived the expression for each of them, which then gives us an expression for the kernel alignment. 2) Since we are interested in the expected alignment over tasks, we take the expectation of the expression found in step 1 over $\beta$ (i.e. the task definition). 3) We write the resulting expression in step 2 in terms of the singular values of the weights, and we show that its optimized when the singular values $s_i$ are distributed evenly across dimensions.
>
> **Better articulation of neuroscience and ML implications**: this is another important comment from the reviewer, and other reviewers have also raised this concern. We apologize for having not made these points clear in our initial submission. In response to this, we have updated Section 4.1 in the manuscript and the paragraph reads as: “*We study the impact of effective weight rank on learning regime due to its potential implications in neuroscience. In particular, learning regimes are indicative of the amount of change undergone through learning, which bear consequences for metabolic costs and catastrophic forgetting (McCloskey & Cohen, 1989; Plaçais & Preat, 2013; Mery & Kawecki, 2005). These considerations are especially pertinent when we examine the various learning regimes evident in neural systems. For instance, during developmental phases, neural systems undergo resource-intensive, plasticity-driven transformations characterized by significant synaptic changes. These transformations are in contrast to the more subdued adjustments observed in mature neural circuits (Lohmann & Kessels, 2014). Based on this understanding, we predict that a circuit's alignment with specific tasks is likely established either through evolutionary processes or during these early developmental phases. As such, the specialization of a neural circuit (e.g., ventral vs dorsal (Bakhtiari et al., 2021)) likely stems from its engagement with tasks that share overlapping computational requirements, ensuring that each circuit is optimally configured for the tasks it is predisposed to perform. Conversely, circuits with high-rank structures, due to their inherent flexibility, are expected to be less specialized, potentially engaging in the learning of a broader range of tasks. With this in mind, our framework could be used as a tool to compare the connectivities across brain regions (and species) to predict their function and flexibility…*”
>
> For deep learning, we note that while low-rank initialization is not common practice, low-rank adaptation and other update parametrizations have recently been popular in large model training schemes. See for example, Low-rank Adaptation (LoRA) in Hu et al. (2021). As such, we argue that the impact of low-rank structures on learning dynamics is quite relevant for current AI practices, and our study contributes to examining how having low-rank structures impact the effective learning regime. The methods and results we present here have the potential to be adapted and built upon to study rankedness and learning regime in several settings, and we are keen to see future work stemming from these ideas. We also note that our study has the tangential benefit of informing the deep learning community about potential applications of their models in other fields such as neuroscience. However, due to the significantly heavier balance of Section 4.1 in discussing neuroscience implication, we have removed “Deep Learning” in the section heading. We have updated Section 4.1 reflecting these points.
>
> **Please continue to the comment below**

---

> > ### Author Response · Authors · 2023-11-16
> > **Response to Reviewer SwWy (2/2)**
> >
> > **Impact on representation learning**: We thank the reviewer for the insightful comment. In line with the reviewer’s suggestion of tracking impact on not just the final trained network but also during the learning process (and related comments by other reviewers), we have now added Appendix Figure 14 tracking the tangent kernel alignment as well as the alignment of the kernel and task throughout training. Figure 14A examines the kernel alignment (alignment of the current kernel to the initial one) in the idealized linear setting with 2D input in Fig. 2 in Atanasov et al. across different initialization schemes: random rank-1 initialization, random full-rank initialization and initially aligned low-rank initialization. We see that low-rank random initializations lead to greater movements, i.e. growing to be more dissimilar to the initial kernel (lower alignment value). Figure 14B looks at the task kernel alignment, $y^T K y / |y|^2 TrK$. Regardless of the initializations, the network kernel moves to be more aligned to the task over training. Low-rank initialization, without being specifically tuned to the task, is the latest to achieve alignment.
> >
> > Figure 14C examines the kernel alignment (alignment of the current kernel to the initial one) in a non-idealized setting: sMNIST task. Again, random low-rank initialization leads to more movement over training, i.e. more dissimilar to the initial kernel. For Figure 14D, with multiple (10) output targets, we track the centered kernel alignment (CKA) of the NTK and class label, e.g. used in Baratin et al. (2021). We again saw that all initializations grow to be more aligned with the class label through training, but higher rank initialization gets to a higher alignment value faster and with less kernel rotation. We have already trained for nearly 20000 SGD iterations and we are not sure if training longer would eventually lead to similar final alignment values across initializations. We note that although achieving higher CKA over training can indicate greater degree of feature learning, a hallmark of the standard rich regime, this does not contradict our findings centered on effective learning regime that is defined based on the amount of change post-training (see Introduction). To incorporate these new findings in the main text, we have added sentences in Simulation Results to bring these Appendix Figures to the attention of an interested reader.

---

> ### Comment · Reviewer_SwWy · 2023-11-20
>
> Thank you to the authors for the detailed responses and thorough revisions.  I agree with the authors and other reviewers that they improve the paper, and I now recommend that the paper be accepted (and am updating my score accordingly).

---

> > ### Author Response · Authors · 2023-11-20
> > **Thank you and further possible improvements to further improve score?**
> >
> > We sincerely appreciate the constructive and insightful feedback provided by this reviewer. We are grateful for the recognition of the improvements made to our revised manuscript. We addressed the major points raised by the reviewer and kindly ask if there are other points we could address for the reviewer to consider increasing their score further beyond the borderline acceptance region.

---

> > > ### Comment · Reviewer_SwWy · 2023-11-20
> > >
> > > I have two remaining concerns:
> > >
> > > -- While I appreciate the authors' additional analysis in Figure 14, the results in Figure 14D appear to be in tension with the intuition underlying Theorem 1 would predict.  Although the authors say that their findings are not contradictory (obviously they are not, assuming all are true!) it makes the findings of the paper harder to interpret for me.  If the authors have a clear understanding of why high-rank initializations result in greater target alignment during learning, despite being overall in a "lazier" regime, an explanation would be very helpful.
> > >
> > > -- I am still not convinced of the concrete applications of this work.  The low-rank adaptation connection seems weak to me -- LoRA uses low-rank weight updates, not low-rank initializations.  The implications for the study of learning and plasticity are rather speculative (perhaps necessarily so) given the limited understanding of the plasticity rules underlying learning in neural circuits.  This point would be strengthened if a concrete, non-obvioius prediction about neural data could be made based on the results of this work (ideally one that is testable given existing data)

---

> ### Author Response · Authors · 2023-11-21
> **Further discussion with Reviewer SwWy (1/2)**
>
> We thank the reviewer for their additional concrete and important comments to further strengthen our manuscript. Please note that we replaced the headings of subsections 4.1 & 4.2, in the previous manuscript version, with bolded text in the updated manuscript to yield extra space for incorporating the discussion points below.
>
> 1) *“While I appreciate the authors' additional analysis in Figure 14, the results in Figure 14D appear to be in tension with the intuition underlying Theorem 1 would predict. Although the authors say that their findings are not contradictory (obviously they are not, assuming all are true!) it makes the findings of the paper harder to interpret for me. If the authors have a clear understanding of why high-rank initializations result in greater target alignment during learning, despite being overall in a "lazier" regime, an explanation would be very helpful.”*
>
> We apologize for not having articulated how this new result integrates into our existing one, and **we hope our intuition below will better clarify and suggest how this new result in Figure 14 is actually in line with the main finding of Theorem 1**. The intuition is in line with the intuition that the reviewer asked us to articulate in their previous comment, *“in high-D spaces, when you draw two random vectors corresponding to the task and the low-rank initialization, they have a high chance of being nearly orthogonal to each other”*. This necessitates more movement for the low-rank initialization to be aligned to the task; consequently, this need for greater movement would require more training (as seen in Figure 14B). If training is stopped early (e.g. due to resource limitations), as likely the case in Figure 14D, they would achieve less final alignment within the training window.
>
> Regarding *“If the authors have a clear understanding of why high-rank initializations result in greater target alignment during learning, despite being overall in a "lazier" regime”*, we would like to remark that cannot be concluded from Figure 14D alone. First of all, these different initializations all lead to similar final alignment in the setting in Figure 14B. Second, we mentioned in our previous response to this reviewer that *“... we are not sure if training longer would eventually lead to similar final alignment values across initializations…”* Based on the upward trend at the end of Figure 14D, we predict that the alignment should continue to rise if we train longer. We note that for sMNIST, we concluded our training upon reaching 97% accuracy, a criteria informed by both published results and our computational resources. We also experimented with halving and doubling the training iterations and observed similar trends.  Whether they would all converge to similar alignment values at the end of training under a wide range of settings ties to one of the proposed future directions (see below).
>
> In response to the reviewer’s feedback, we have updated the caption of Figure 14 clarifying these points. Also, since this discussion is in line with our proposed future direction on deeper investigation of the link between rankedness, the learning regime and the consequent impact on representation (including kernel task alignment) and generalization properties of converged solutions under a wide range of settings, we have updated the corresponding sentence in Discussion alerting the reader to Appendix Figure 14.
>
> **Please continue to the comment below**

---

> ### Author Response · Authors · 2023-11-21
> **Further discussion with Reviewer SwWy (2/2)**
>
> 2) *“I am still not convinced of the concrete applications of this work...”*
>
> Although LoRA focuses on parameter updates rather than initializations, the exploration of how the rank of these updates influences learning regimes is a crucial area for future research. Our results provide an initial toolkit that can potentially be adapted for different contexts (e.g. with low-rank updates and structure enforced).
>
> While different rank initializations, to our knowledge, have been sparsely examined (an example work can be found in Vodrahalli et al. (2022)), our study indicates that they should be examined more. Indeed, our result shows that the learning regime can be influenced by initialization rank (or other structure that has an impact on rank), and that the learning regime can significantly impact the nature of the solutions learned, specifically with respect to generalization (George et al., 2022). Consequently, it's conceivable that inductive biases in the form of ranked initialization could be employed to influence the generalization characteristics of neural networks, an aspect we have identified as a potential direction for future research. We thank the reviewer for pressing us on this point, which we recognize to be important for the ML community. As such, we add this discussion point to our discussion section.
>
> For concrete neuroscience predictions, continuing the previously added/modified discussion paragraph mentioned in the previous response, *“... With this in mind, our framework could be used as a tool to compare the connectivities across brain regions (and species) to predict their function and flexibility ”*, we predict that the effective rank of a circuit, as determined from connectomic datasets, could indicate its level of functional specialization. Low-rank connectivity suggests a higher learning cost for diverse tasks, potentially rendering these circuits less adaptable for generalist roles. However, we acknowledge that the link between rankedness and specialization could be confounded by other factors — such as the underlying plasticity rule (also recognized by the reviewer) and the task rank that have been mentioned in Discussion, underscoring the need for deeper explorations in the future.
>
> An additional concrete prediction of our theory would be that for neural circuits learning a new task, connectivity rank would influence how much change in task-dependent neural activity is expected before vs after training. Existing brain-machine interface experimental protocols already allow to explore this question, by measuring changes in activity during learning following readout perturbation, which was done by substituting a new mapping connecting
> neural activity to BCI output positions (i.e. re-initializing the readout weights). In previous work (Sadtler et al. (2014) and Golub et al. (2018)), the authors show that learning following a perturbation differs whether or not the perturbation was within neural activity manifold, or outside of it, necessitating some realignment to the task. One could apply this analysis to test the prediction by examining how the learning differs depending on the rank of the new readout. We have updated Discussion reflecting these points.
>
> Overall, we note that the study of learning regimes is of great interest to the neuroscience community as it can inform ways by which neural circuits may exploit synaptic plasticity to perform credit assignment. A number of recent publications explore this topic, including Bordelon and Pehlevan (2023) that explores the impact of biologically plausible learning rule on learning dynamics and representation in different learning regimes, and Flesch et al. (2022) that compare experimental data to expected neural activity changes in distinct regimes. Overall, much like for the deep learning advantages outlined above, there is an interesting question about the characteristics of solutions that are discovered by different learning regimes, with some being more prone to spurious correlations and therefore, with distinct generalization properties (George et al., 2022) to be further examined in the future. It is likely that different brain regions employ different learning regime strategies and our contribution helps the community effort in building a toolbox to explore this question.
>
> In summary, we hope we convinced the reviewer that the question of learning regime and connectivity is of great interest from different perspectives, and that our results help contribute to better understanding impacts on both biology and AI.

---

### Official Review · Reviewer_Cn5Y · 2023-10-31

**Soundness:** 3 good
**Presentation:** 3 good
**Contribution:** 3 good
**Rating:** 8
**Confidence:** 4

**Summary:**

This study examines how the initial connectivity of a neural network affects its learning regime (lazy vs rich). In particular. the authors focus on the rank of the connectivity, in contrast to e.g. weight variance studied in previous work. The main motivation for this is the claim that biological networks have low-rank connectivity. Both in theory using feedforward linear networks, and in experiments using recurrent non-linear networks, it is shown that low-rank initialisation lead to richer learning than full-rank initialisation (except, if the low-rank initialisation aligns with the task). This is intuitive as high rank initialisation means it is likely that some (linear combination of the) columns of weight matrix are already aligned with the task matrix, whereas a low-rank matrix will most likely be orthogonal with the task at initialisation.

**Strengths:**

The question of how connectivity influences representation and learning dynamics is of significance, both for the neuroscientists, as well as for machine learning scientists. How low-rank initialisation affects the learning regime has as far as I am aware not been studied in depth, yet networks with low-rank weight matrices are used by both the deep-learning and computational neuroscience communities.

The paper is well written and the main idea is clearly presented. Multiple experiments are performed that highlight the main result and the supplementary contains a large amounts of controls. The experimental results are supported by theoretical results on linear feedforward networks.

Overall I lean on accepting this paper, and I would increase my score if the concerns below are addressed

**Weaknesses:**

1. While many controls are done, the initial dynamic regime of the network (stable, unstable or chaotic) is not controlled for (see questions below).
2. I am not entirely convinced by the evidence given for the statement that the brain has low-rank connectivity. The paper repeatedly cites Song et al., 2005 and Mastrogiuseppe & Ostojic, 2018 as evidence for low-rank connectivity in the brain. The latter is a purely theoretical study, which I am not sure why it used as evidence for biology. The former indeed studies biological neural circuits, and found that the distribution of synaptic connection strength can be fitted by a lognormal distribution, as well as the circuits having overrepresented bidirectional connectives. While my intuition is that this implies low-rank connectivity, it can (and likely should) be made clear if this is the case. In general the relation between local connectivity patterns and (global) rank is not always obvious a priori (potentially, see Shao & Ostojic 2023).
3. Biological constraints are only used as initialisation and not enforced during training. This decreases the relation of the simulations to biology, as neurons in the brain can (generally) not just decide to forget about e.g. Dale’s law during learning.
4. The authors claim to study continuous-time networks, however, $dt$ is set equal to $\tau_m$, and tasks have around 10 time steps after discretisation. It could be questioned how well the simulations reflect the continuous-time equations.

Minor:
1. Equation 10, last line should say d instead of 3 above the equality sign.
2. Figure 3B left, title is cut off.
3. The 5th references seems formatted wrong.
4. Equation 1. -> It could be named explicit that you are making an exponential Euler approximation (which is not used in the studies cited here).

**Questions:**

The questions are all related to the initial dynamic regime versus initial connectivity

1.  I would like to the authors to comment on whether the difference in lazy versus rich learning is potentially confounded by the regime of the initial dynamics (stable, unstable or chaotic), instead of the rank of the initial matrix. Three results that lead me to hypothesise there is at least some influence of dynamic regime on the learning:

- For Figure 1, the large full rank networks will likely be initially be chaotic, given a gain of 1.5 (transition at $\sqrt(2)$ for ReLU networks; Kadmon & Somplinsky 2015). However at least part of the truncated networks (despite rescaling by the norm) will likely be not.
- The outlier eigenvalue in the networks of Figure 2 might lead to (unstable) dynamics initially exploding in the direction of the corresponding eigenvector, instead of following a chaotic regime.
- Finally, in Figure 9, the differences between low-rank and full-rank networks diminishes when both are initialised in the stable regime.

2. Were the Dale’s law networks balanced? E.g. increasing inhibitory strength if less then 50% of neurons are inhibitory, so that the expected input to neurons stays zero, would put the Dale’s law networks in similar regimes as the controls (see Rajan & Abbott, 2006).

3. The Eigenspectrum, and as a result the initial dynamics, of the full-rank random networks are most likely approximately constant between networks, whereas the (non-zero part of the) eigenspectrum of the low-rank networks will vary more over initialisations - is this related to the observed high variance in richer learning between the low-rank models?

Note that getting to same dynamics could be partly ameliorated by scaling by the largest singular value instead of the norm (which are realistically only approximately proportional for the large full rank networks). However this also is not a guarantee, as e.g. as stated above balancing also has an influence on dynamics

---

> ### Author Response · Authors · 2023-11-16
> **Response to Reviewer Cn5Y (1/2)**
>
> We thank the reviewer for their insightful contextualization of this work, effective articulation of our intuition and providing constructive and executable suggestions to both improve the content and presentation of this work. Below, we explain how we address all of the reviewer’s concerns.
>
> **Confounding factor from dynamical regime**: We thank the reviewer for this insightful comment/question. Indeed, it is plausible that the learning regime in RNNs could be influenced by the initial dynamical regime. With stable dynamics (non chaotic) , for instance, activity typically settles into steady states in absence of inputs over time steps, potentially needing more weight adjustments during training to effectively propagate information to the last time step. The study of learning dynamics and dynamical regime of RNNs is rich and involved and there are multiple ways to control for dynamics stability. A common method is through the leading weight eigenvalue of the connectivity matrix, which directly influences the top Lyapunov exponent dictating the rate of trajectory expansion.
>
> As such, we now add details about a new control where we compare the leading weight eigenvalue (Appendix Figure 19), and we observed a consistent trend: higher-rank initializations lead to effectively lazier learning post-training. We note that a deeper exploration of the relationship between learning regime and various notions of the dynamical regime is a promising avenue for future work but falls outside the scope of the present paper, and we express this view in the discussion. We thank the reviewer for this insightful comment/question, and in response, we have updated the manuscript that added sentences in Simulation Results to reflect these points and alert interested readers to Appendix Figure 18.
>
> **Evidence that the brain uses low-rank connectivity**: we thank the reviewer for pointing out this important and subtle point. In response to the reviewer’s important point, we have toned down our claim throughout the manuscript (Abstract, Introduction and Discussion in particular): there are evidence (e.g. Song et al.) \textbf{suggestive} of effective low-rank structures in the brain, rather than hard evidence. We also removed the citation to Mastrogiuseppe & Ostojic here, as the reviewer pointed out. We originally had a sentence in between the two citations but it got lost in the editing.
>
> The extent to which the brain utilizes low-rank structures remains an open question, given the significant variation in neural circuit structures across regions and species. On top of that, as the reviewer pointed out, while local connectivity statistics can offer some predictive insight into the global low-rank structure, this relationship is not always immediately apparent (Shao & Ostojic, 2023). Our theoretical results contribute to this discourse by providing tools that link connectivity features, particularly effective rank, with learning (see Section 4.1). We have updated the manuscript reflecting this discussion in Related Works and Limitations & Future Directions.
>
> **Please continue to the comment below**

---

> ### Author Response · Authors · 2023-11-16
> **Response to Reviewer Cn5Y (2/2)**
>
> **Biological constraints enforced during training**: we agree with the reviewer that enforcing constraints (e.g. Dale’s law) would be important for biological relevance. We initially didn’t do this to test our theoretical prediction, which assumes gradient-descent update without constraints. Following the reviewer’s comment, we have added a new figure (see Appendix Figure 17) with Dale’s law enforced for the EI experiments and observed similar trend as before: EI initialization led to greater changes post-training. We have also added sentences in Simulation Results alerting readers to this appendix figure.
>
> **Continuous RNN settings**: we thank the reviewer for this important comment. We have now added Appendix Figure 18 to show our main conclusions — that higher rank random initializations lead to effectively lazier learning — also hold for the case when $\tau_m > dt$ and with longer sequence length. Specifically, we kept $\tau_m=100$ but changed $dt$ from $100$ to $20$, which increased the sequence length by a factor of 5. We have also added sentences in Simulation Results alerting readers to this appendix figure.
>
> **Balance Dale’s Law**: we thank the reviewer for this question and we apologize that it wasn’t clear. We used 80% excitatory and indeed balanced the initialization according to standard practices. We have updated Setup & Simulation Details reflecting this point.
>
> **High variance due to variable eigenspectrum in low-rank network**: However, our newly added Appendix Figure 19, which maintains a relatively stable leading eigenvalue by fixing the dominant eigenvalue across comparisons, still results in high variance for low-rank initializations. This suggests that there may be other factors contributing to this high variance, and it would be interesting to examine this in the future. This point is also reflected in the sentences added to Simulation Results, as a part of responses to the reviewer's earlier comment about the dynamical regime as a possible confounding factor.
>
> **Minor fixes**: we thank the reviewer for catching these and we have fixed them.

---

> > ### Comment · Reviewer_Cn5Y · 2023-11-17
> > **Response to revisions**
> >
> > First of all, I want to applaud the authors for such a thorough response! I have increase my score, as the following points are now all addressed:
> >
> > - Whether or not the brain has low-rank connectivity is now phrased in a more appropriate manner
> >
> > - The case of constraints not being applied during training, and the case of large time integration steps are clearly resolved.
> >
> > - The control analysis (Figure 19), largely ameliorates the issue of the initial dynamic regime. As noted above this does not 100% fix the issue, but this is also clearly acknowledged in the paper.
> >
> >
> > I found two typos:
> >
> > - Page 6: archtiecture -> architecture
> >
> > - Figure 15 Caption:  A reference is broken

---

> > > ### Author Response · Authors · 2023-11-17
> > > **Thank you**
> > >
> > > We sincerely thank the reviewer for their crucial feedback and appreciate their recognition of our thorough response, as well as their thoughtful consideration in adjusting our score. Additionally, we have corrected the typos identified by the reviewer.

---

### Official Review · Reviewer_RNHT · 2023-11-02

**Soundness:** 3 good
**Presentation:** 2 fair
**Contribution:** 2 fair
**Rating:** 5
**Confidence:** 3

**Summary:**

This paper studies how structure in the weight matrices -- focusing on the recurrent weight matrices in RNNs, and feedforward matrix in two layer linear feedforward networks -- shapes rich and lazy learning (roughly whether the weights change a lot or negligibly through learning respectively). The paper finds that high-rank initializations lead to so-called "lazier" learning, whereas lower rank initializations tend to lead to "richer" learning, or larger changes in the weights. If the low-rank intialization is already suitable for the task, there can be negligible changes to the weights as well.

**Strengths:**

The results (analysis in linear feedforward networks, and experiments in RNNs) appear technically correct, and are in line with existing literature on how weights change during learning. Note that I skimmed through the Appendix and did not closely check the proofs.

**Weaknesses:**

I felt like the paper could provide a more finer-grained description and insight into how the initialization scheme affects the final solution. This could be by providing insight into how the existing metrics (from Sect 2.3) were changing during learning as a function of training epoch (rather than just applied on the final solution, esp. when in the rich regime) and/or through the use of additional finer-grained metrics. As it stands, it feels like there could be different explanations for how the weights are evolving during learning (that I think are interesting and important); for example is the largest rank component changing first or changing in magnitude the most during learning? Does this happen even if the task is low-rank but the (low rank) initialization is not perfectly aligned with the task?

I think this could be addressed both in experiments as well as in the linear setting where the authors make analytical statements. For example, Prop 1 appears considers a low-rank 1 initialization completely aligned with the task (initialized with $\beta$) -- but I think it would be helpful to understand in general how does a low rank initialization compare with a high rank initialization for learning some fixed (low rank) task, rather than an expectation over all tasks? In other words in general would a low-rank initialization be better for a low rank task and why/why not? also how are weights/singular components evolving to facilitate the learning?

As an aside, the paper could be made stronger if the case that the theory studied was closer to the experiments (though I understand that it is easier to analyze in the linear FF setting and that the theory still provided insight for the RNN experiments).

The presentation and writing was generally clear but can be improved. While reading the paper, it felt like, at times, the writing was too verbose and could have been made clearer. For example, "A nexus between deep learning and neuroscience has expanded the applications of deep learning theoretical frameworks to study the learning dynamics of biological neural network", and " Our work builds on these studies by further exploring the precursors of these regime" could be more clear.  Minor: Fig 3B, left title not visible

**Questions:**

- For the proof in the Appendix of Thm 1, while do you need the scale $\sigma = \||W_i(0)||_F$ to be small? Also small relative to what?
- Is there evidence that the initial connectivity structures in biology are low-rank?

---

> ### Author Response · Authors · 2023-11-16
> **Response to Reviewer RNHT (1/2)**
>
> We thank the reviewer for their accurate summary of our work. We also thank the reviewer for their insightful comments to enrich the study and concrete suggestions to improve the presentation. We explain below how we address the reviewer’s comments.
>
> **More fine-grained description of the impact of initialization schemes**: In line with the reviewer’s suggestion of tracking impact on not just the final trained network but also during the learning process (and related comments by other reviewers), we have now added Appendix Figure 14 tracking the tangent kernel alignment as well as the alignment of the kernel and task throughout training. Figure 14A examines the kernel alignment (alignment of the current kernel to the initial one) in the idealized linear setting with 2D input in Fig. 2 in Atanasov et al. across different initialization schemes: random rank-1 initialization, random full-rank initialization and initially aligned low-rank initialization. We see that low-rank random initializations lead to greater movements, i.e. growing to be more dissimilar to the initial kernel (lower alignment value). Figure 14B looks at the task kernel alignment, $y^T K y / |y|^2 TrK$. Regardless of the initializations, the network kernel moves to be more aligned to the task over training. Low-rank initialization, without being specifically tuned to the task, is the latest to achieve alignment.
>
> Figure 14C examines the kernel alignment (alignment of the current kernel to the initial one) in a non-idealized setting: sMNIST task. Again, random low-rank initialization leads to more movement over training, i.e. more dissimilar to the initial kernel. For Figure 14D, with multiple (10) output targets, we track the centered kernel alignment (CKA) of the NTK and class label, e.g. used in Baratin et al. (2021). We again saw that all initializations grow to be more aligned with the class label through training, but higher rank initialization gets to a higher alignment value faster and with less kernel rotation. We have already trained for nearly 20000 SGD iterations and we are not sure if training longer would eventually lead to similar final alignment values across initializations. We note that although achieving higher CKA over training can indicate greater degree of feature learning, a hallmark of the standard rich regime, this does not contradict our findings centered on effective learning regime that is defined based on the amount of change post-training (see Introduction).
>
> In addition, to examine how the leading component changes relative to the rest, we track the evolution of the kernel effective rank, inspired by Baratin et al. (2021) in Appendix Figure 15. Here, the effective rank is the ratio of the trace to the dominant one, thereby capturing how much the leading eigenvalue evolves relative to the rest. We found that the kernel effective rank is getting closer to the target/task dimension over training.
>
> Please note that we focused the additional analyses on the network kernel rather than individual layer weights, as the former is more reflective of the network’s function as a whole. To incorporate these new findings in the main text, we have added sentences in Simulation Results to bring these Appendix Figures to the attention of an interested reader.
>
> **Feedforward theory and RNN experiments**: We agree with the reviewer’s comment. Note that the new addition of feedforward experiments in non-idealized setting (in response to Reviewer rK41) would be a step closer to what the reviewer suggests about closer theory and experiment setting.
>
> **Improving presentation**: We thank the reviewer for this suggestion. We fixed several more verbose sentences in addition to the one the reviewer pointed out, including the very first sentence in Introduction. We have also fixed the typo that the reviewer pointed out for Figure 3B (now in Figure 2) in the original submission.
>
> **Please continue to the comment below**

---

> > ### Author Response · Authors · 2023-11-16
> > **Response to Reviewer RNHT (2/2)**
> >
> > **Helpful to understand low vs high rank initialization for learning some fixed (low rank) task, rather than an expectation over all tasks? Would a low-rank initialization be better for a low rank task and why/why not?** We thank the reviewer for these questions and we apologize for any confusions we caused. Before directly answering the reviewer’s question, we would first like to clarify that in our RNN experiments (e.g. 2AF), we examined a fixed task (although there is some degree of stochasticity in data presentation due to SGD). Our theoretical result — focusing on the expectation — primarily aims to explain the randomness/uncertainty arising from random initialization that has no knowledge of the specific/fixed task prior to training. Importantly, looking at the expectation across tasks also gives some insights into how flexible the circuit is when learning across a broad range of tasks.
> >
> > Regarding the question of whether low-rank initialization would help for a low-rank task, we again remark that all the neuroscience tasks studied here are known to be low-rank. Our idealized linear feedforward task, accompanying Theorem 1, is also low-rank. As such, this discussion would connect back to the gist of our results: without having prior knowledge about the task, i.e. without explicitly aligning the circuit, random low-rank initialization would require greater adjustments (or harder time) to learn these tasks (even if these tasks are low-rank). These low-rank structures would mainly be helpful (requiring less changes) if they were aligned to the task statistics prior to training (the aligned initialization results). We also remark that more nuanced interplay of learning dynamics across different initialization rank and task dimension is left for future work, as mentioned in Discussion. We have also updated the Discussion section in response to the reviewer’s great question.
> >
> > **Small \sigma relative to what**: we thank the reviewer for this question. Here, small $\sigma$ means $\sigma<<1$. This small initial weight variance assumption allows us to use  t the analytical expression ofr the tangent kernel after training derived by Atanasov et al., which is valid up to terms that are at least quadratic in  $\sigma$. We have updated the manuscript clarifying this point.
> >
> > **Evidence for low-rank structure in neural circuits**: We appreciate the reviewer's great question. In response to this reviewer and reviewer Cn5Y's comments, we have added an overview of the implications of local connectivity structures (in both Related Works as well as Limitations and Future Work), as evidenced in works like Song et al. (2005), and their potential influence on global low-rankedness, for example, as discussed in Shao & Ostojic (2023). While local connectivity statistics can indeed provide some predictive insight into global low-rank structures, the relationship is not always immediately obvious. Additionally, the degree to which the brain employs low-rank structures is still an open question, considering the vast variability in neural circuit structures across different regions and species. Consequently, we have moderated our claims regarding the low-rankedness in neural circuits throughout our manuscript (Abstract, Introduction and Discussion in particular). Importantly, we believe our theoretical findings significantly contribute to this ongoing discourse by offering predictions that connect specific connectivity features, especially effective rank, with learning (see Section 4.1). We have updated the manuscript reflecting this point in Limitations and Future Work.

---

> ### Author Response · Authors · 2023-11-21
> **Request for discussion**
>
> Dear Reviewer RNHT,
>
> As the author-reviewer discussion period nears its end, we would appreciate any feedback on whether our responses have addressed your concerns. If there are additional questions or points we can address, please let us know. We hope our recent updates have resolved the initial concerns and are committed to further improving our paper through this constructive dialogue.
>
> Thank you, Authors

---

> ### Comment · Reviewer_RNHT · 2023-11-22
>
> I would like to thank the authors for the detailed reply. I have also read the other reviewer comments. I think the presentation and clarity has been improved in the revised manuscript. I also appreciate that the authors have done a large and commendable amount of experiments including the additional ones during the rebuttal period, which have addressed some of my concerns.
>
> However, I still am of the sentiment (similar to reviewer SwWy) that the paper does not clearly explain *why* low rank initializations lead to "richer" learning. For example, this finding appears to be dependent on the choice of the initial gain $g$ in the recurrent weight matrix (in Fig 8d in Appendix, for gain < 1 weights appears to change more with increasing rank), which suggests there is a more nuanced explanation than only the rank of the initial weight matrix. This also appears to differ to the  feedforward theory which was true when $\sigma << 1$, which also sets the scale at initialization. Related (and minor): it would be also helpful to clarify the details of the initialization for the new feedforward experiments done in the revision.

---

> > ### Author Response · Authors · 2023-11-22
> > **Further discussion with Reviewer RNHT**
> >
> > We thank the reviewer for their careful reading of our updated manuscript and acknowledgement of our detailed reply, as well as their astute question. As per this reviewer’s comment, we have now added the simulation details for the new feedforward experiments in Appendix C. We also appreciate this reviewer’s reference to Reviewer SwWy's inquiry about understanding the link between low-rank initializations and effectively richer learning; in response to Reviewer SwYy's valuable point, we have provided further clarification and intuition in Section 2.3, following Theorem 1, in our rebuttal revision.
> >
> > In response to *“... which suggests there is a more nuanced explanation than only the rank of the initial weight matrix”*, the reviewer is absolutely correct that there are other factors, besides the initial weight rank, that can confound the learning regime. For instance, as cited in the manuscript, many previous studies have examined the influence of initial hidden weight magnitude (Flesch et al., 2022; Chizat et al., 2019), the so-called $\alpha$ parameter (Chizat et al., 2019; George et al., 2022; Bordelon et al., 2022) on learning regimes. There are additional factors; notably, the dynamical regime for RNNs — which can be influenced by the initial recurrent weight gain noted by the reviewer — could also confound the learning regime (also mentioned in our discussion with Reviewer Cn5Y). This rapidly growing literature showcases both the acute interest of the field for this topic, and the breadth of factors to influence learning regimes. As such, like in all citations outlined above, assumptions and choices of factors to explore need to be made in order to properly frame progress. Overall, we agree with the reviewer that there are remaining questions regarding mechanisms with respect to the interaction between rank and other regime factors. However, we believe these questions are part of an important direction the field is actively exploring, and cannot all be addressed in one paper. In our case, we believe we do properly motivate our choices of investigative assumptions, properly control for confounding factors and, crucially, discuss the critical directions our work opens up for future work. Below, we outline our reasoning with respect to the important points raised by the reviewer.
> >
> > In our manuscript, we strive to explore the effect of rank on effective learning regime, and attempt to isolate this connectivity feature. With extensive control experiments (also acknowledged by Reviewer Cn5Y), we show that despite other factors (e.g. network size) influencing learning regime, rank has general effects that endures across a wide range of configurations. This is a crucial control for which we thank the reviewers to push us to communicate. We agree with the reviewer that our results do not offer a complete understanding of this mechanism for RNNs, but we argue that exploring several mechanisms at once is beyond the scope of a single paper but that our results are an important step toward this goal. In respond to the reviewer’s feedback, we have added a sentence to the discussion paragraph in the manuscript — where we discussed confounding factors and motivating further exploration of additional confounding factors (e.g. noise) in the future — and the sentence says *“Examining several mechanisms at once is beyond the scope of one paper, but our theoretical work constitutes the foundation for future investigations”*. Motivated by the reviewer’s response, we have also added the sentence *“Overall, this dynamic area of learning regimes is ripe for many explorations, integrating numerous factors; our work contributes to this exciting area with new tools”* to the end of Discussion in the updated manuscript.
> >
> > To summarize, we acknowledge the reviewer’s position about remaining open questions about learning regimes and RNNs. But, our study enables these questions to be asked more precisely. However, due to the large number of possible things that compound effects on learning regimes, we believe it is necessary to concentrate on specific pointed mechanisms from a theoretical perspective, while of course controlling for others. We do this in our paper, proposing Theorems based on the feedforward setting, and then motivating and verifying their validity in recurrent networks, while experimentally controlling for confounding factors and highlighting important future work directions. As such, we believe our paper brings new contributions to the field, and that this warrants consideration for publication at ICLR.

---

### Author Response · Authors · 2023-11-16
**General Response**

We are grateful to all four reviewers for their highly engaged, insightful and concrete feedback which helped us greatly improve our manuscript. In our revised submission, which we have detailed below, we have worked to **address all the concerns** raised and incorporated suggestions to further illuminate the key points for our readers. As recommended, we have expanded our discussion to more explicitly connect our results to their implications in neuroscience. We have incorporated various reviewer’s suggestions to further improve the clarity and fix potentially misleading sentences. Additionally, new simulations have been included to show our trends across broader settings (see Appendix Figures 16-19 and revised main text that points to it) and to examine the nuanced impacts of different initializations during learning (instead of just at the end), particularly in terms of kernel alignment (Appendix Figure 14). Changes in the updated manuscript are highlighted in \textcolor{blue}{blue} in the revised pdf for ease of reference.

We are encouraged by the reviewers' recognition of our work's technical correctness (e.g. Reviewer RNHT). Reviewer Cn5Y highlighted the significance of our work for both neuroscience and machine learning communities, particularly noting the novelty of our exploration into how low-rank initialization affects learning regimes. Reviewer SwWy commended the clarity of our writing and the solid evidence provided by our experiments supporting our main takeaway that lower-rank initializations generally lead to richer learning dynamics. Lastly, Reviewer rK41 found our paper timely and the core idea novel and interesting.

Overall, we believe these enhancements significantly improve both the content and presentation of our submission, and it is our opinion that the scientific community would benefit from a timely communication of these results as it is central to a rapidly expanding body of work on learning regimes in connection to both neuroscience and AI.

---

### Meta-Review · Area_Chair_RPmQ · 2023-12-03

**Metareview:**

The paper investigates the effect of the distribution of initial weights on the learning dynamics of neural networks, with a focus on recurrent neural networks, tasks and connectivity structures often used in neuroscience studies. Using a combination of simulations and theoretical analysis (of simplified models), they investigate how the effective rank of a neural network determines whether networks will learn in a ‘lazy’ or ‘rich’ manner. Reviewers generally appreciated the study, although there were some concerns that the paper was insufficiently nuanced and clear about which aspects of the network and connectivity are critical for determining the learning regime.

**Justification For Why Not Higher Score:**

I do think this is a fine ICLR paper, but I do share the reviewers view that the paper likely takes a bit of a simplistic view about what influences learning dynamics, and I would be hesitant to nominate it for an award/oral.

**Justification For Why Not Lower Score:**

All reviewers recommended acceptance, and I see no critical overlooked flaw that would make me overturn their consensus.

---

### Decision · Program_Chairs · 2024-01-16

Accept (poster)